JCB Journal of Cell Biology

# CDC42EP5/BORG3 modulates SEPT9 to promote actomyosin function, migration, and invasion

Aaron J. Farrugia[1], Javier Rodríguez[2], Jose L. Orgaz[3], María Lucas[2], Victoria Sanz-Moreno[3], and Fernando Calvo[1,2]

Fast amoeboid migration is critical for developmental processes and can be hijacked by cancer cells to enhance metastatic dissemination. This migratory behavior is tightly controlled by high levels of actomyosin contractility, but how it is coupled to other cytoskeletal components is poorly understood. Septins are increasingly recognized as novel cytoskeletal components, but details on their regulation and contribution to migration are lacking. Here, we show that the septin regulator Cdc42EP5 is consistently required for amoeboid melanoma cells to invade and migrate into collagen-rich matrices and locally invade and disseminate in vivo. Cdc42EP5 associates with actin structures, leading to increased actomyosin contractility and amoeboid migration. Cdc42EP5 affects these functions through SEPT9-dependent F-actin cross-linking, which enables the generation of F-actin bundles required for the sustained stabilization of highly contractile actomyosin structures. This study provides evidence that Cdc42EP5 is a regulator of cancer cell motility that coordinates actin and septin networks and describes a unique role for SEPT9 in melanoma invasion and metastasis.

## Introduction

Malignant melanoma is a very aggressive type of skin cancer due to its highly metastatic behavior, which relies on the increased ability of melanoma cells to migrate and invade (Lo and Fisher, 2014). Melanoma cells can migrate as single cells that display two major morphologies: elongated mesenchymal or rounded amoeboid. Mesenchymal migration is characterized by Rac-driven actin-based protrusions, matrix degradation, and strong focal adhesions (FAs) coupled to actin fibers that enable transmission of forces (Pandya et al., 2017). In contrast, amoeboid migration modes are characterized by a rounded morphology as well as blebs, lower levels of adhesion, and high levels of actomyosin contractility (Paluch et al., 2016). Amoeboid migration plays important roles in developmental processes and immune cell function (Madsen and Sahai, 2010; Richardson and Lehmann, 2010). Additionally, amoeboid behavior is prominent in the invasive fronts of melanomas in animal models (Herraiz et al., 2015; Sanz-Moreno et al., 2008, 2011) and human lesions (Georgouli et al., 2019; Orgaz et al., 2014; Sanz-Moreno et al., 2011). It has also been associated with increased risk of metastasis and poorer prognosis (Georgouli et al., 2019), which underlies the need for a better mechanistic understanding of the process. Actomyosin contractility driven by the motor protein myosin II is critical for rounded migration (Tozluoğlu et al., 2013). This process has been shown to be tightly controlled by Rho-ROCK signaling leading to increased phosphorylation of the

regulatory myosin light chain 2 (MLC2; Vicente-Manzanares et al., 2009). However, how actin structures are organized and coordinated with other cytoskeletal components to enable their correct assembly and the formation of fully functional actomyosin networks is not well understood.

Septins are a large conserved family of GTP-binding proteins that participate in a broad spectrum of cellular functions (Mostowy and Cossart, 2012). Septins have been proposed as the fourth component of the cytoskeleton due to their ability to form higher-order structures such as filaments, which can associate with distinct subsets of actin filaments and microtubules, as well as membranes of specific curvature and composition (Spiliotis, 2018). Importantly, septins are emerging as crucial regulators of the generation, maintenance, and positioning of cytoskeletal networks with potential roles in cell migration. In line with this, different septins have been shown to be required for mesenchymal migration in epithelial and endothelial cells (Dolat et al., 2014; Liu et al., 2014). In addition, septins form a uniform network at the cell cortex in leukocytes, and SEPT7 expression is required for rapid cortical contraction during dynamic shape changes (Gilden et al., 2012; Tooley et al., 2009). In cancer, a potential role for septins in modulating aggressiveness is also starting to emerge (Angelis and Spiliotis, 2016; Poüs et al., 2016), although little is known about the molecular details and functions of individual members in melanoma and amoeboid migration.

[1]Division of Cancer Biology, The Institute of Cancer Research, London, UK; [2]Instituto de Biomedicina y Biotecnología de Cantabria (Consejo Superior de Investigaciones Científicas, Universidad de Cantabria), Santander, Spain; [3]Centre for Cancer and Inflammation, Barts Cancer Institute, Queen Mary University of London, London, UK.

Correspondence to Fernando Calvo: calvof@unican.es.

Although a role for septins in modulating cytoskeletal rearrangements is clearly emerging, the regulatory mechanisms required for establishing and maintaining these interactions in migratory cells are still elusive. Binder of Rho GTPases (Borg) proteins (also called Cdc42 effector proteins [Cdc42EPs]) are among the few proteins known to interact with septins and regulate their function (Farrugia and Calvo, 2016). Borg proteins vary in length, but all contain a Borg homology 3 domain (BD3) that binds septins (Farrugia and Calvo, 2016). Although they remain largely uncharacterized, recent studies suggest crucial roles of Borg proteins in regulating cytoskeletal organization and related cellular processes (Calvo et al., 2015; Liu et al., 2014). Yet the exact role of Borg proteins and septins in migration and their relationship in modulating cancer cell invasion and metastatic dissemination remain to be determined.

Here, we investigate the contribution of individual Borg proteins to rounded-amoeboid cell migration and invasion using melanoma models. We find that Cdc42EP5 is consistently required for these processes in vitro and for local invasion and metastasis in vivo. Mechanistically, we demonstrate that Cdc42EP5 acts by inducing actin cytoskeleton rearrangements that potentiate actomyosin function in a septin-dependent manner, and we show a unique role for SEPT9 in controlling actomyosin contractility, migration, and invasion.

## Results

### Identification of CDC42EP5 as a regulator of melanoma migration, invasion, and metastasis

To assess the relevance of Borg proteins Cdc42ep1–Cdc42ep5 in cancer cell migration and invasion, we first analyzed transwell migration after RNAi silencing of individual genes in the murine melanoma model 690.cl2 (Dhomen et al., 2009; Kümper et al., 2016). Note that when referring to murine RNA/proteins, we use an uppercase letter followed by all lowercase letters (i.e., Cdc42ep5 or abbreviated forms [Ep5]), and when referring to human products or in general, we use uppercase letters (i.e., CDC42EP5 or EP5). We achieved knockdown of all Borg genes when each gene was specifically targeted (Fig. S1 A). Fig. 1 A shows that knocking down Cdc42ep3 and Cdc42ep5 significantly reduced the ability of 690.cl2 cells to migrate through transwell pores, whereas silencing Cdc42ep2 increased migration. To assess how these defects were affecting the ability of melanoma cells to invade, we employed collagen invasion assays and observed that only Cdc42ep5 depletion significantly reduced 690.cl2 invasion (Fig. 1 B). Silencing Cdc42ep5 expression with two independent RNAis in 690.cl2 cells (Fig. 1 C) yielded similar results (Fig. 1, D and E). Using CRISPR-Cas9 technology (Fig. S1, B and C), we generated Cdc42ep5-knockout (KO) 690.cl2 cells expressing GFP (690.cl2$^{KO}$-GFP) and knockout cells ectopically reexpressing an N-terminal GFP-Cdc42ep5 fusion (690.cl2$^{KO}$-GFP-Cdc42ep5). Critically, GFP-Cdc42ep5 expression in the reconstituted 690.cl2$^{KO}$ cells was similar to endogenous Cdc42ep5 levels in parental 690.cl2 cells. Functional characterization informed that Cdc42ep5 depletion affected melanoma migration in vitro and that these defects were abrogated when Cdc42ep5 expression was reconstituted (Fig. S1 D).

There was also a significant increase in the migratory abilities of parental 690.cl2 cells after Cdc42ep5 overexpression (Fig. S1 E), confirming a role for this protein in melanoma migration and invasion in vitro. These results were further validated in an alternative human melanoma model (WM266.4) and a model of metastatic breast cancer (MDA-MB-231-LM2), where CDC42EP5 was consistently required for migration and invasion (Fig. S1, F–H). We observed that Cdc42ep5/CDC42EP5 expression was consistently low when compared with other Borg genes in all these systems (Fig. S1, I and J), indicating that its relevance is not a result of increased levels or abundance.

Melanoma cells can invade locally and disseminate to distant organs including the lungs (Calvo et al., 2011). To confirm the requirement for CDC42EP5 in these processes in vivo, we first investigated lung metastasis after tail vein injection of control and Cdc42ep5-knockdown 690.cl2 cells in mice (Fig. 1 F). 2 h after injection, equal numbers of control and Cdc42ep5-depleted cells lodged in the lungs (Fig. 1 G), indicating that Cdc42ep5 did not affect survival in circulation. However, after 24 h, the number of Cdc42ep5-knockdown cells that invaded into the lung parenchyma was significantly reduced when compared with control cells, confirming a role for Cdc42ep5 in metastatic colonization in vivo. Next, we investigated melanoma motility and local invasion using intravital imaging in living tumors (Fig. 1 H). For this long-term assay, we employed the CRISPR-engineered cell lines 690.cl2$^{KO}$-GFP and 690.cl2$^{KO}$-GFP-Cdc42ep5, which will allow for assessing the specific role of Cdc42ep5 and discard clonal effects derived from knockout generation. No changes in melanoma proliferation or tumor growth were observed after modulating Cdc42ep5 expression (Fig. S1, K and L). Intravital imaging showed an increase in the speed of moving cells in 690.cl2$^{KO}$-GFP-Cdc42ep5 cells when compared with 690.cl2$^{KO}$-GFP cells (Fig. 1, I and J; and Videos 1 and 2). Together, these results indicate that CDC42EP5 is consistently required for melanoma migration and invasion in vitro and for local invasion and metastatic dissemination in vivo.

### CDC42EP5 promotes actomyosin function in collagen-rich matrices

Fast motility of melanoma cells in vivo has been associated with a particular rounded-amoeboid behavior that promotes local invasion and metastasis, negatively impacting patient survival (Sanz-Moreno et al., 2008; Sanz-Moreno and Marshall, 2010). In melanoma, amoeboid migration in vivo is tightly regulated by actomyosin function (Herraiz et al., 2015; Sanz-Moreno et al., 2008, 2011). Importantly, rounded-amoeboid behavior in melanoma can be modeled in vitro by seeding cells on collagen-rich matrices and assessing cell morphology and actomyosin function (Sahai and Marshall, 2003; Sanz-Moreno et al., 2008).

To explore whether CDC42EP5 was promoting migration and invasion by participating in these processes, we first used confocal microscopy to determine the precise localization of Cdc42ep5 with respect to actomyosin networks in 690.cl2 cells seeded on collagen-rich matrices. We could not identify suitable Cdc42ep5 antibodies for immunofluorescence, so we stably expressed our GFP-Cdc42ep5 construct in 690.cl2 cells

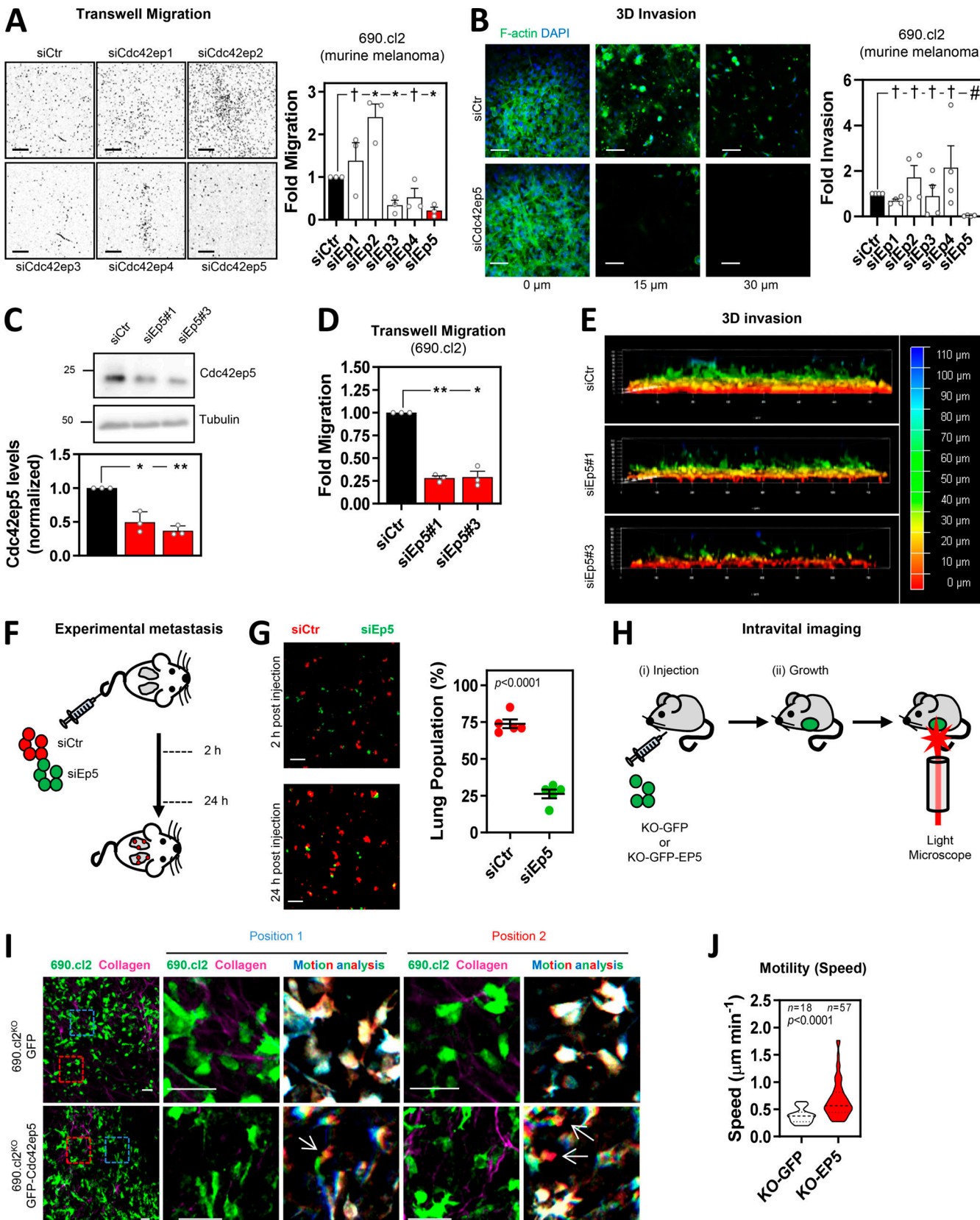

Figure 1. **Directed siRNA screen identifies CDC42EP5 as a new regulator of melanoma migration and invasion. (A)** Images show DAPI staining of migrated 690.cl2 murine melanoma after transfection with control (siCtr) or Cdc42ep1-5 (siEp1-5) siRNA. Scale bars, 1 mm. Graph shows fold migration. Bars indicate mean ± SEM (*n* = 3 independent experiments; one-way paired ANOVA, Dunnett's test: †, not significant; *, P < 0.05). **(B)** Images show F-actin (green) and DAPI (blue) staining of 690.cl2 cells at different levels of collagen invasion after transfection with indicated siRNAs. Scale bars, 100 μm. Graph shows fold

invasion. Bars indicate mean ± SEM ($n$ = 3 independent experiments; one-way paired ANOVA, Dunnett's test: †, not significant; #, P < 0.0001). **(C)** Representative Western blot showing indicated protein levels in 690.cl2 cells after transfection with control (siCtr) or two individual Cdc42ep5 (siEp5) siRNAs. Graph shows quantification of Cdc42ep5 protein levels. Bars indicate mean ± SEM ($n$ = 3 experiments; one-way paired ANOVA, Dunnett's test; *, P < 0.05; **, P < 0.01). **(D)** Fold migration of 690.cl2 cells after transfection with indicated siRNAs. Bars indicate mean ± SEM ($n$ = 3 independent experiments; one-way paired ANOVA, Dunnett's test: *, P < 0.05; **, P < 0.01). **(E)** Images show 3D reconstructions of collagen invasion assays of 690.cl2 cells transfected with indicated siRNAs. Color scale bar indicates depth of invasion. **(F)** Experimental metastasis approach. Equal numbers of control (siCtr, red) and Cdc42ep5-knockdown (siEp5, green) 690.cl2 cells were injected in the tail vein of mice. 2 and 24 h after injection, lungs were analyzed by microscopy. **(G)** Images of mouse lungs at 2 and 24 h after tail injection with 690.cl2 cells as in F. Scale bars, 100 µm. Graph shows proportions (%) of siCtr and siEp5 cells in lungs at 24 h. Lines represent mean ± SEM ($n$ = 5 experiments; unpaired $t$ test). **(H)** Intravital imaging approach. 690.cl2$^{KO}$-GFP or 690.cl2$^{KO}$-GFP-Cdc42ep5 cells were injected subcutaneously in mice. Grown living tumors on anaesthetized mice were then exposed and imaged using two-photon microscopy. **(I)** Intravital imaging of 690.cl2$^{KO}$-GFP or 690.cl2$^{KO}$-GFP-Cdc42ep5 tumors (green, melanoma cells; magenta, collagen fibers). Right panels show motion analysis images from different time points where static cells appear white; arrows indicate fast-moving cells represented as a trail of blue-green-red pseudocolors. Scale bars, 50 µm. **(J)** Violin plot showing individual speed of moving cells from I. $n$, moving cells in five mice per condition; Mann–Whitney test. KO, knockout.

to investigate its cellular localization. Initial analyses informed that individual cells presented different degrees of Cdc42ep5 expression, which were positively correlated with actomyosin function (as measured by pS19-MLC2 intensity levels), F-actin levels, and cell roundness, suggesting a potential role for Cdc42ep5 in regulating these parameters (Fig. S2, A and B). In-depth confocal analysis of Cdc42ep5-expressing rounded 690.cl2 cells on collagen-rich matrices informed that GFP-Cdc42ep5 localized preferentially at the cell cortex (Fig. 2 A), colocalizing with two key components of cortical actomyosin networks, F-actin and pS19-MLC2.

A role for CDC42EP5 in modulating actomyosin contractility in 3D was further confirmed by perturbation analyses. Thus, silencing Cdc42ep5 with two independent RNAis significantly decreased the roundness index of 690.cl2 cells seeded on collagen-rich matrices (Fig. 2, B and C; and Fig. S2, C and D). Importantly, the reduction in cell rounding after Cdc42ep5 depletion was associated with a decrease in F-actin and pS19-MLC2 levels by immunofluorescence (Fig. 2, B and C). Immunoblot analyses of whole-cell lysates confirmed the reduction of pS19-MLC2, but not total MLC2, levels after Cdc42ep5 silencing (Fig. 2 D). As a result, the ability of cells to contract collagen-rich matrices was severely impeded after Cdc42ep5 depletion (Fig. 2 E), which underlines the relevance of Cdc42ep5 in modulating actomyosin activity and cellular contractility. The association between cell rounding and CDC42EP5 expression was consistent in other melanoma cell lines. Thus, we found that *CDC42EP5* was the only Borg gene whose expression was significantly correlated with cell roundness (as assessed on cells seeded on collagen-rich matrices) in a panel of 11 human melanoma cell lines of varying degrees of rounding (Orgaz et al., 2014; Figs. 2 F and S2 E). Perturbation analyses confirmed this observation, as silencing Borg genes *CDC42EP1–CDC42EP4* in 690.cl2 had no effect on pS19-MLC2 levels, which contrasted with the significant decrease after *CDC42EP5* depletion (Fig. S2 F). Together, these data indicated that CDC42EP5 promotes the generation of an actomyosin cortex, which impacts actomyosin activity and cell morphology, and induces invasive phenotypes in melanoma cells in 3D settings.

### CDC42EP5 regulates the organization of actomyosin networks and is required for the maturation of FAs

While actomyosin contractility in 3D pliable environments drives rounded-amoeboid behavior, in stiff undeformable 2D culture, it can be associated with the formation of stress fibers, long actin filaments decorated with active MLC2 (pS19-MLC2; Pellegrin and Mellor, 2007). To assess if actomyosin was regulated by Cdc42ep5 independently of the physical context, we next used 2D culture. We observed that Cdc42ep5 formed an intricate filamentous network that overlapped fibrillar actin bundles in the perinuclear region in 690.cl2 cells (Fig. 3 A). Cdc42ep5 filaments were less evident toward the cell periphery. Time-lapse imaging showed that Cdc42ep5 filaments were dynamic and aligned with F-actin fibers, stretching along the cell body toward the cell periphery (Video 3). Cdc42ep5 filaments were absent from lamellipodial regions and membrane ruffles. In agreement, Pearson's correlation analyses informed that Cdc42ep5 colocalized with F-actin preferentially in the perinuclear region in fixed cells (Fig. 3, A and B). There was no colocalization of GFP-Cdc42ep5 with acetylated α-tubulin; therefore, direct associations between Cdc42ep5 and microtubules were unlikely (Figs. S3 A and 3 B). More detailed analyses of cellular structures in 2D informed that GFP-Cdc42ep5 filaments aligned with both F-actin– and pS19-MLC2–positive filaments (i.e., stress fibers; Fig. S3 B). In cellular protrusions, Cdc42ep5 localized with actomyosin filaments mostly around the edges of the protrusion. Confocal analysis of the perinuclear region indicated that Cdc42ep5 filaments stretched across the basal side in alignment with F-actin fibers, with activated MLC2 dotted around the network. Cdc42ep5 also localized in the apical region, where it formed a "wavy" network around actomyosin filaments.

To establish a causal connection between Cdc42ep5 filaments and actomyosin structures in 2D, Cdc42ep5 expression was silenced in 690.cl2 cells using two independent RNAis. Cdc42ep5 knockdown severely affected F-actin organization, as actin stress fibers were completely disrupted, especially at the perinuclear region (Fig. 3 C). In contrast, the meshwork pattern of fibrillar actin at the cell periphery was not affected, although actin in the lamellipodial areas appeared thinner and more condensed. This result showed that Cdc42ep5 was particularly required for stress fiber stabilization. Importantly, CDC42EP5-associated phenotypes were extensible to other cell types, as similar results were obtained after depletion of CDC42EP5 in human melanoma cells (WM266.4) and in murine embryonic fibroblasts (Fig. S3 C). These data suggest that CDC42EP5 is a conserved regulator of the actomyosin cytoskeleton.

The actomyosin complex is critical for the growth and elongation of cell–ECM adhesions required for efficient cell

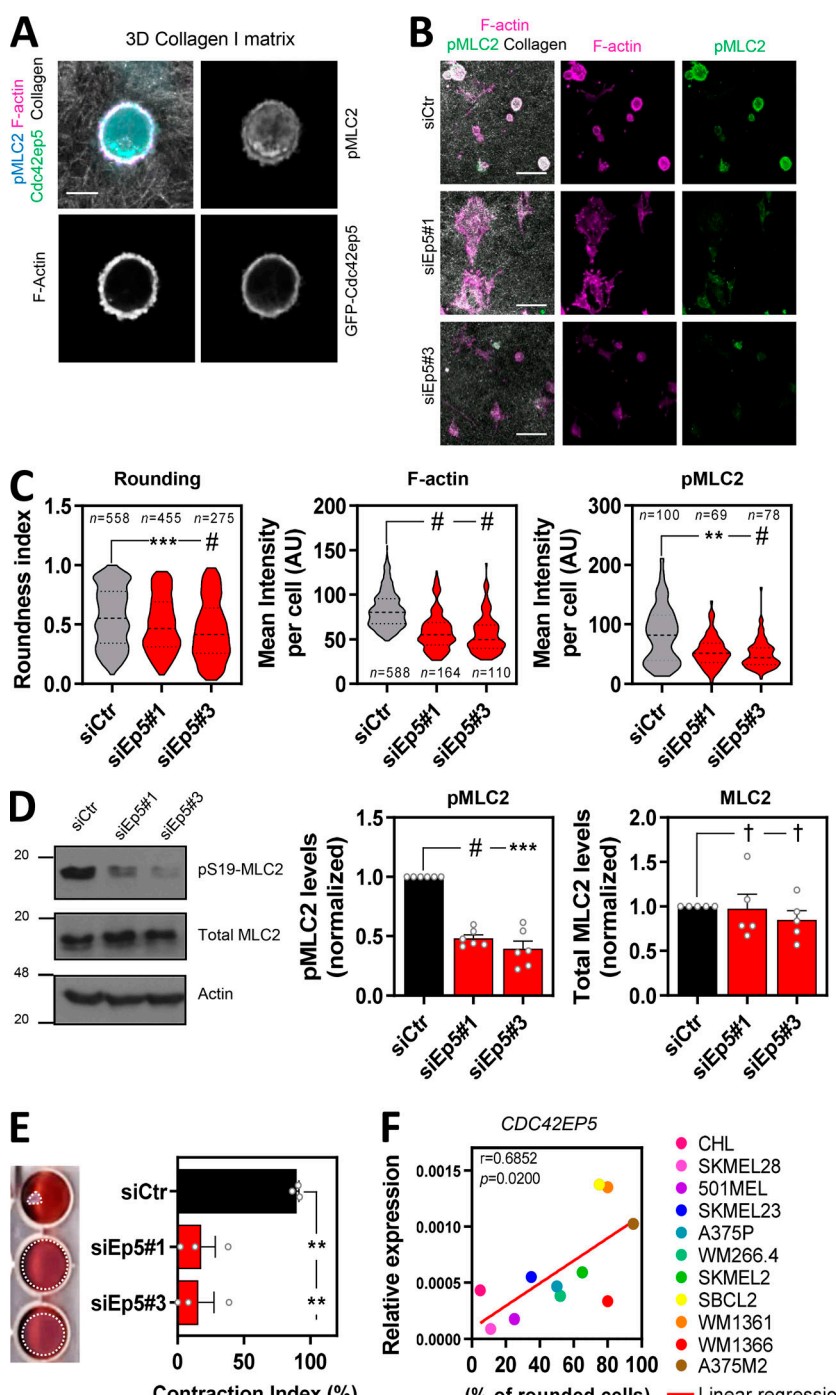

Figure 2. **CDC42EP5 modulates actomyosin function in melanoma in 3D. (A)** Image of a 690.cl2 cell on collagen-rich matrix expressing GFP-Cdc42ep5 (green) and stained for F-actin (magenta) and pS19-MLC2 (cyan); collagen fibers (gray) were captured by reflectance signal. Individual grayscale channels are also shown. Scale bar, 10 µm. **(B)** Images of 690.cl2 cells on collagen-rich matrices after transfection with control (siCtr) or two individual Cdc42ep5 (siEp5) siRNAs. Images show staining for F-actin (magenta) and pS19-MLC2 (green); collagen fibers (gray) were captured by reflectance signal. Scale bars, 40 µm. **(C)** Violin plots showing roundness index (left), F-actin mean intensity (middle), and pS19-MLC2 mean intensity (right) of individual cells from B. $n$ represents individual cells; Kruskal–Wallis and Dunn's tests: **, P < 0.01; ***, P < 0.001; #, P < 0.0001. **(D)** Western blots showing indicated protein levels in 690.cl2 cells after transfection with control (siCtr) and two individual Cdc42ep5 (siEp5) siRNAs. Graphs show quantification of normalized pS19-MLC2 and total MLC2 levels. Bars indicate mean ± SEM ($n$ = 6 [pMLC2] and 5 [MLC2] independent experiments; one-way paired ANOVA, Dunnett's test: †, not significant; ***, P < 0.001; #, P < 0.0001). **(E)** Images and graph showing gel contraction of 690.cl2 cells after indicated treatments. Bars indicate mean ± SEM ($n$ = 3 independent experiments; one-way paired ANOVA, Dunnett's test: **, P < 0.01). **(F)** Graph shows expression of *CDC42EP5* mRNA normalized to *GAPDH* expression in human melanoma cell lines with increasing rounding coefficients. Pearson's correlation coefficient ($r$), statistical significance ($p$), and linear regression (red line) are shown. Each point in the graph represents the mean value of three independent experiments.

migration, as it mediates the maturation of nascent focal complexes to FAs (Burridge and Wittchen, 2013). This process requires myosin-mediated tension as well as the formation and maintenance of an F-actin network (Choi et al., 2008). The protein Paxillin is recruited early into focal complexes and plays a key scaffolding role at FAs. Its phosphorylation at Y31 and Y118 is crucial for FA formation (Petit and Thiery, 2000). Our previous observations suggested that CDC42EP5 may be part of the actomyosin machinery that drives the maturation of focal complexes to FAs in melanoma. To test this in more detail, we first investigated Cdc42ep5 association with FAs. Confocal

microscopy analyses showed that Cdc42ep5 was present at actin fibers that connected with FAs but showed no clear localization at pY188-Paxillin areas (Fig. 3 D). Together, these data indicated that Cdc42ep5 is not part of the FA but localizes at high contractile F-actin filaments associated with FAs, suggesting a potential role in FA maturation. To test this, FA formation and dynamics in 690.cl2 cells were studied after Cdc42ep5 silencing. Using pY118-Paxillin staining, we observed that Cdc42ep5-depleted cells presented smaller and dot-like adhesions that localized closer to the cell periphery compared with control cells (Fig. 3 E), features that are characteristic of

Figure 3. **Cdc42ep5 promotes stress fiber formation and FA maturation in 2D. (A)** Image shows a 690.cl2 cell expressing GFP-Cdc42ep5 (green) on glass and stained for F-actin (magenta) and DAPI (blue). Additional panels show magnifications of perinuclear and peripheral areas. Scale bars, 12.5 µm. **(B)** Graph shows Pearson's correlation coefficient of GFP-Cdc42ep5 signal against perinuclear and peripheral F-actin staining and tubulin staining. Bars indicate mean ± SEM

(n = 10 independent cells [perinuclear and peripheral F-actin] and 5 [tubulin]; one-way paired ANOVA, Dunnett's test: *, P < 0.05; ***, P < 0.001; #, P < 0.0001. **(C)** Images of 690.cl2 cells after transfection with control (siCtr) and two individual Cdc42ep5 (siEp5) siRNAs on glass. Images show F-actin (magenta) and DAPI (blue) staining with zoom-up panels showing perinuclear and peripheral regions. Scale bars, 12.5 μm. Graph is a violin-plot quantification of F-actin intensity in the perinuclear and peripheral regions in the indicated points. n, specific regions in individual cells; Kruskal–Wallis and Dunn's tests: *, P < 0.05; #, P < 0.0001. **(D)** Images show individual and merged channels of 690.cl2 cells expressing GFP-Cdc42ep5 (green) on glass and stained for F-actin (magenta) and pY118-Paxillin (blue). Scale bar, 5 μm. Graph at the bottom is a violin plot showing the Pearson's correlation coefficient of Cdc42ep5 and F-actin staining against pY118-Paxillin–positive areas. (n, pY118-Paxillin–positive areas in individual cells; unpaired t test.) **(E)** Images of 690.cl2 cells on glass. Images show F-actin (magenta), pY118-Paxillin (green), and DAPI (blue) staining. Single-channel magnification of pY118-Paxillin staining is also shown. Scale bars, 25 μm. Graph is a violin plot showing quantification of FA size (i.e., pY118-Paxillin–positive area). n, peripheral regions in individual cells; Kruskal–Wallis and Dunn's: #, P < 0.0001. **(F)** Images show motion analysis of Paxillin-GFP after TIRF imaging. Scale bars, 25 μm. **(G)** Western blots showing indicated protein levels in 690.cl2 cells after transfection with control (siCtr) and two individual Cdc42ep5 (siEp5) siRNAs. Right graphs show quantification of normalized p418-Src levels. Bars indicate mean ± SEM (n = 5 independent experiments; one-way paired ANOVA, Dunnett's test: *, P < 0.05; ***, P < 0.001).

immature focal complexes (Petit and Thiery, 2000). Zyxin is a FA protein that distinguishes mature FA from focal complexes, as it is recruited later into the complex (Petit and Thiery, 2000). Knockdown of Cdc42ep5 in 690.cl2 cells resulted in a complete loss of Zyxin-positive adhesions (Fig. S3 D). Importantly, this defect was observed when cells were cultured both on glass and on fibronectin, ruling out any potential effect associated with defective cell adhesion. Furthermore, time-lapse imaging of Paxillin-GFP showed that Cdc42ep5 knockdown affected the elongation of FAs, which appeared static over time (Video 4 and Fig. 3 F). As a result of deficient FA maturation, Cdc42ep5-depleted cells failed to activate FA downstream signaling (i.e., Src; Calvo et al., 2015; Fig. 3 G) and presented defects in single-cell migration in 2D (Fig. S3 E). Interestingly, it was observed that Cdc42ep5 was the main Borg protein involved in the process. While silencing Cdc42ep1 and Cdc42ep3 marginally reduced pY418-Src levels, knocking down Cdc42ep5 resulted in significant inhibition of Src activity (Fig. S3 F). Together, these data indicated that in 2D melanoma cells CDC42EP5 forms filamentous structures that overlap actin filaments and promotes the formation of perinuclear actomyosin fibers, leading to the maturation of FAs and increased cell motility.

## SEPT9 promotes actomyosin activity, invasion, and metastasis in melanoma

Borg proteins, including CDC42EP5, have been shown to primarily promote septin filament assembly (Calvo et al., 2015; Joberty et al., 2001). In humans, septins are encoded by 13 different genes that encode multiple septin isoforms (Mostowy and Cossart, 2012). Septins can associate with distinct actin networks, modulate their formation, and facilitate myosin activation (Spiliotis, 2018). In amoeboid T lymphocytes, it has been proposed that septins tune actomyosin forces during motility (Gilden et al., 2012). To assess whether CDC42EP5 was functioning via septins to regulate actomyosin function and invasion in melanoma, we first analyzed whether there was any particular septin associated with melanoma aggressiveness. Analyses of datasets of human material (Riker et al., 2008; Talantov et al., 2005) indicated that *SEPT9* was the only septin gene whose upregulation is associated with melanoma progression and dissemination (Fig. 4 A; and Fig. S4, A and B). These results pointed to a specific role of SEPT9 in melanoma invasion and metastasis that we sought to investigate further.

SEPT9 is the only member of the septin family that directly promotes actin polymerization and cross-linking (Dolat et al.,

2014; Smith et al., 2015). However, other septins such as SEPT2 and SEPT7 have been shown to colocalize with actin fibers and modulate their formation in nonmalignant cells such as fibroblasts (Calvo et al., 2015), lymphocytes (Gilden et al., 2012) and endothelial cells (Liu et al., 2014), as well as breast cancer cells (Zhang et al., 2016). To assess SEPT9 relevance in melanoma in comparison to other critical septins (i.e., SEPT2 and SEPT7), we proceeded to analyze the effect of perturbing Sept2, Sept7, and Sept9 function in 690.cl2 cells by specifically targeting their expression by RNAi. All individual treatments specifically decreased the expression of their targeted septin, although there was a slight reduction in Sept9 expression after depletion of Sept7 (Figs. 4 B and S4 C). Only disruption of Sept9 expression affected both cell roundness and the generation of an actomyosin cortex (as read by pS19-MLC2 and F-actin levels) in 690.cl2 cells seeded on collagen-rich matrices (Fig. 4, C and D; and Fig. S4 D). Similar findings were observed after SEPT9 depletion in human melanoma WM266.4 cells (Fig. S4 E). Analyses of cells in 2D culture showed that only depletion of Sept9 and, to a lesser extent, Sept7 affected the generation of perinuclear F-actin fibers (Fig. 4 E). Similar to Cdc42ep5 depletion, silencing Sept9 led to a marked reduction in FA size (Fig. 4 E). Sept2-depleted cells also exhibited smaller FAs. In support of a specific role of Sept9 in controlling actomyosin contractility in melanoma, we observed that Sept9 silencing significantly reduced the ability of parental 690.cl2 cells to migrate in transwell assays (Fig. 4 F). There was no significant change in the migration ability after Sept2 or Sept7 knockdown. In 3D collagen invasion assays, knockdown of Sept9 resulted in a significant decrease in invasion (Fig. 4 G). On the other hand, Sept2 depletion did not affect invasion whereas silencing Sept7 surprisingly increased the invasive potential of 690.cl2 cells. Importantly, these phenotypic and functional defects were associated with a reduced ability of 690.cl2 cells to metastasize in tail-vein assays after Sept9 depletion (Fig. 4 H). Altogether, these data point to SEPT9 as the key regulator of the actomyosin cytoskeleton during melanoma local invasion and distant metastatic colonization.

## CDC42EP5 promotes SEPT9 actin cross-linking, leading to enhanced actomyosin activity

Having established a specific role for SEPT9 in regulating actomyosin contractility and invasion in melanoma, we next sought to investigate whether CDC42EP5 was modulating SEPT9 function in these processes. 690cl2 cells in 2D cultures presented

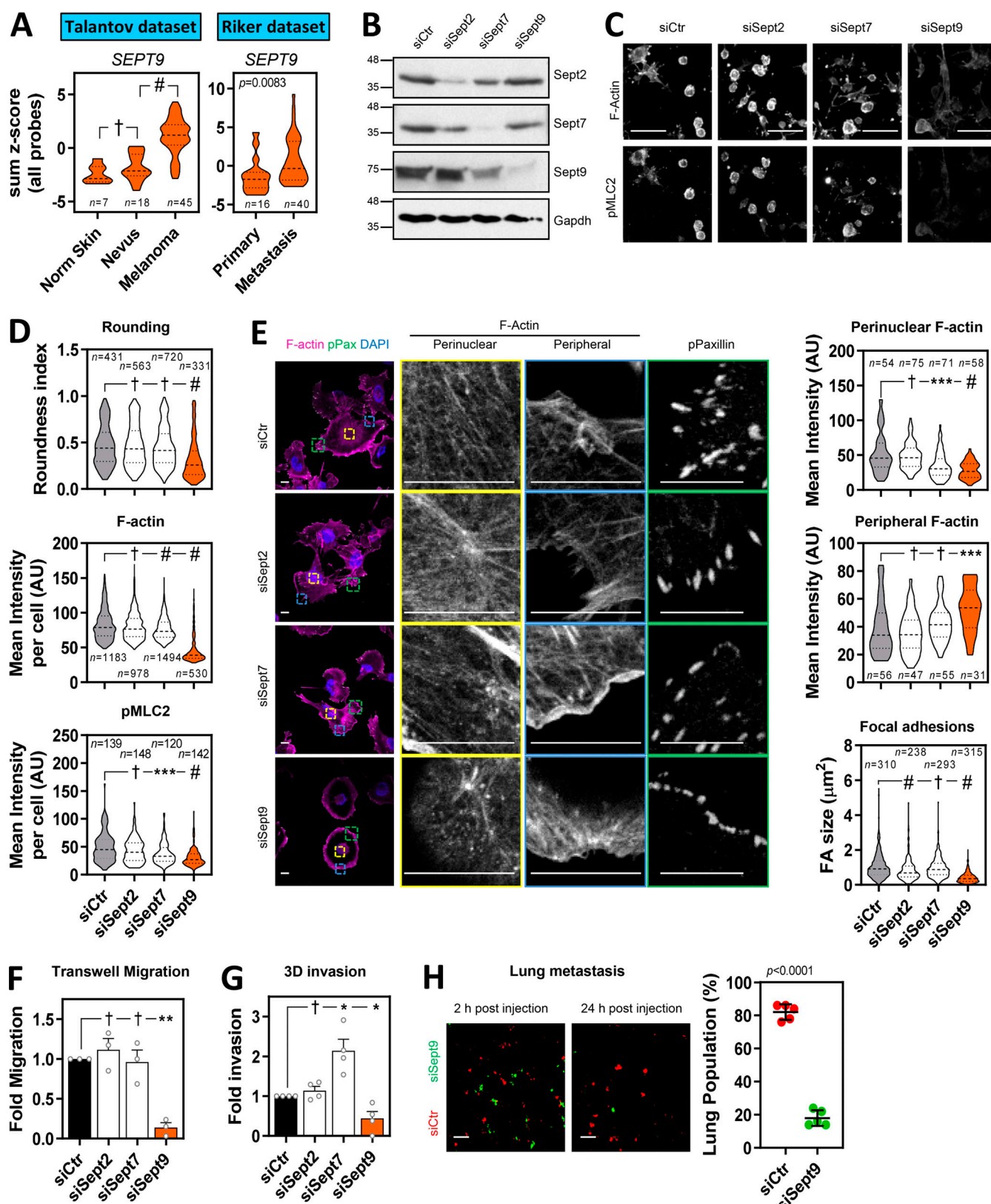

Figure 4. **SEPT9 is required for actomyosin contractility and invasion in melanoma. (A)** Violin plots show *SEPT9* expression in human tissues from normal (Norm) skin, nevus, and melanoma (left: Talantov et al. dataset, GSE3189) and primary melanoma and metastatic melanoma (right: Riker et al. dataset, GSE7553). *n*, individual patients; Kruskal–Wallis and Dunn's tests [Talantov] and Mann–Whitney test [Riker]: †, not significant; #, P < 0.0001. **(B)** Western blots showing indicated protein levels in 690.cl2 cells after transfection with control (siCtr) and Sept2, Sept7, and Sept9 siRNA. **(C)** Images show F-actin and pS19-MLC2 staining of parental 690.cl2 after transfection with control (siCtr), Sept2, Sept7, and Sept9 siRNA on collagen-rich matrices. Scale bars, 75 μm. **(D)** Violin

plots showing roundness index (top), F-actin mean intensity (center), and pS19-MLC2 mean intensity (bottom) of individual cells from C. *n*, individual cells; Kruskal–Wallis and Dunn's tests: †, not significant; ***, P < 0.001; #, P < 0.0001. **(E)** Images show F-actin (magenta), pY118-Paxillin (green), and DAPI (cyan) staining in 690.cl2 cells after transfection with indicated siRNAs on glass. Right panels show indicated magnifications of F-actin staining in the perinuclear and peripheral regions, and pY118-Paxillin staining at the cell border. Scale bars, 12.5 µm. Violin plots show quantification of F-actin intensity in the perinuclear (top) and peripheral (center) regions, and quantification of FA size (bottom). *n*, single regions from individual cells [top and center] and individual FAs [bottom]; Kruskal–Wallis and Dunn's tests: †, not significant; ***, P < 0.001; #, P < 0.0001. **(F and G)** Graphs show fold migration ability (F) and fold invasion into collagen (G) of 690.cl2 cells after transfection with indicated siRNAs. Bars indicate mean ± SEM (*n* = 3 [F] and 4 [G] independent experiments; paired *t* tests: †, not significant; *, P < 0.05; **, P < 0.01). **(H)** Images of mouse lungs at 2 and 24 h after tail injection with 690.cl2 cells transfected with control (siCtr-Red) or Sept9 (siSept9-Green) siRNA. Scale bars, 100 µm. Graph shows proportions (%) of siCtr and siSept9 cells within the lungs at 24 h. Lines represent mean ± SEM (*n* = 5 mice; unpaired *t* test). AU, arbitrary units.

Cdc42ep5 colocalized with Sept9 filaments (Fig. 5 A). These filaments exhibited a clear overlap with F-actin stress fibers in the perinuclear region, but not in the cell periphery. Similarly, Sept9 was also observed in the actin cortex with Cdc42ep5 and F-actin in amoeboid 690.cl2 cells seeded on collagen-rich 3D matrices (Fig. 5 B). Importantly, modulating Cdc42ep5 expression amply affected the formation of Sept9 structures. Thus, we observed that silencing Cdc42ep5 reduced the formation of perinuclear Sept9 filaments and increased its cytosolic pool in 2D culture (Fig. 5 C). In 3D environments, Sept9 presented a sparse cytosolic pattern in 690.cl2[KO]-GFP, whereas stable reconstitution of Cdc42ep5 expression redistributed Sept9 to cortical regions cells (Fig. 5 D).

As opposed to other septins, SEPT9 is the only known septin that binds actin filaments directly and can promote the cross-linking of prepolymerized actin filaments, even at suboptimal concentrations (Dolat et al., 2014; Smith et al., 2015; Fig. S5 A). Using low-speed actin sedimentation assays, we confirmed that neither Cdc42ep5 nor SEPT6/7 induced F-actin cross-linking on their own (Fig. S5 B). Interestingly, we observed that addition of recombinant Cdc42ep5 increased the amount of cross-linked F-actin induced by SEPT9 (Fig. 5 E). In addition, using F-actin polymerization assays and F-actin–binding assays, we confirmed previous findings describing the ability of recombinant SEPT9 to bind F-actin and promote actin polymerization (Dolat et al., 2014; Smith et al., 2015; Fig. S5, C and D); on the other hand, Cdc42EP5 exhibited no actin polymerization activities and a minimal interaction with F-actin.

### SEPT9 is a crucial effector of CDC42EP5 function in melanoma

These data suggested that CDC42EP5 acted primarily by reinforcing the ability of SEPT9 to cross-link actin filaments into stress fibers and cortical actin, which is required for actomyosin contractility in 2D and 3D settings, respectively. To further validate this idea, we generated a Cdc42ep5 mutant defective in interaction with Sept2/6/7 by mutating key residues in the BD3 domain of Cdc42ep5 (Cdc42ep5[GPS-AAA] mutant; Fig. 6 A; Calvo et al., 2015; Joberty et al., 2001). We confirmed that Cdc42ep5 was also able to interact with Sept9 and that this interaction depended on an intact BD3 domain (Fig. S6 A).We assessed the activity of this mutant in comparison to wild-type Cdc42ep5 (Cdc42ep5[WT]) by gain-of-function analyses in 690.cl2[KO] cells (Fig. S6 B). Contrary to Cdc42ep5[WT] reconstitution, expression of Cdc42ep5[GPS-AAA] in 690.cl2[KO] cells in 2D did not induce the formation of perinuclear Sept9 networks (Fig. S6 C), actin stress fibers or matured FAs (Fig. 6 B). In 3D, expression of Cdc42ep5[GPS-AAA] in 690.cl2[KO] cells did not affect cell rounding and pS19-MLC2 levels, whereas Cdc42ep5[WT] significantly increased these parameters (Fig. 6 C; Fig. S6, D and E). Importantly, these phenotypes were concomitant with functional defects, as septin binding was absolutely required for Cdc42ep5-dependent transwell migration and collagen invasion (Fig. 6, D and E). Overall, these results confirm that Cdc42ep5 requires Sept9 interactions to promote actomyosin-dependent proinvasive behaviors in melanoma.

To further confirm that SEPT9 was a crucial effector of CDC42EP5 function in melanoma, we next investigated whether elevating Sept9 activity was sufficient to rescue the functional defects associated with loss of Cdc42ep5. For this, we used a SEPT9 isoform containing the N-terminal domain (SEPT9_V1), which has been shown to present enhanced activities (Gonzalez et al., 2007). Thus, reconstituting SEPT9 activity in 690.cl2[KO] cells by ectopic expression of SEPT9_V1 was able to induce cell rounding and actomyosin activity in 3D culture (Figs. 6 F and S6 F). In addition, SEPT9_V1 expression reconstituted perinuclear actin fiber and FA formation in 690.cl2[KO] cells (Fig. 6 G) and increased their migratory capabilities to levels similar to those obtained by Cdc42ep5 expression (Fig. 6 H). Overall these results illustrate a unique role for Sept9 in controlling actomyosin structure and function in melanoma and place it as the key effector of CDC42EP5 functions in regulating migration, invasion, and metastasis.

## Discussion

Here, we identify CDC42EP5 as a new regulator of melanoma invasion and metastasis. We demonstrate that CDC42EP5 forms filamentous structures that associate with F-actin and promotes the assembly of higher-order actomyosin bundles in 2D and 3D. Contrary to other actin bundling regulators such as Fascin, α-Actinin, and Filamin (Stevenson et al., 2012), CDC42EP5 promotes actin bundling not directly but by controlling septin network reorganization. Thus, a septin-binding defective mutant is not able to affect actin structures or confer migratory and invasive capabilities in melanoma.

Septins such as SEPT2, SEPT4, SEPT6, SEPT7, and SEPT9 have been shown to colocalize with actin most prominently in stress fibers, and their perturbation can alter actin organization (Spiliotis, 2018). In particular, SEPT7 is proposed to be the essential septin in the formation of septin oligomers and filamentous structures (Zent et al., 2011). In addition, SEPT2 binds myosin II directly and this interaction is proposed to promote

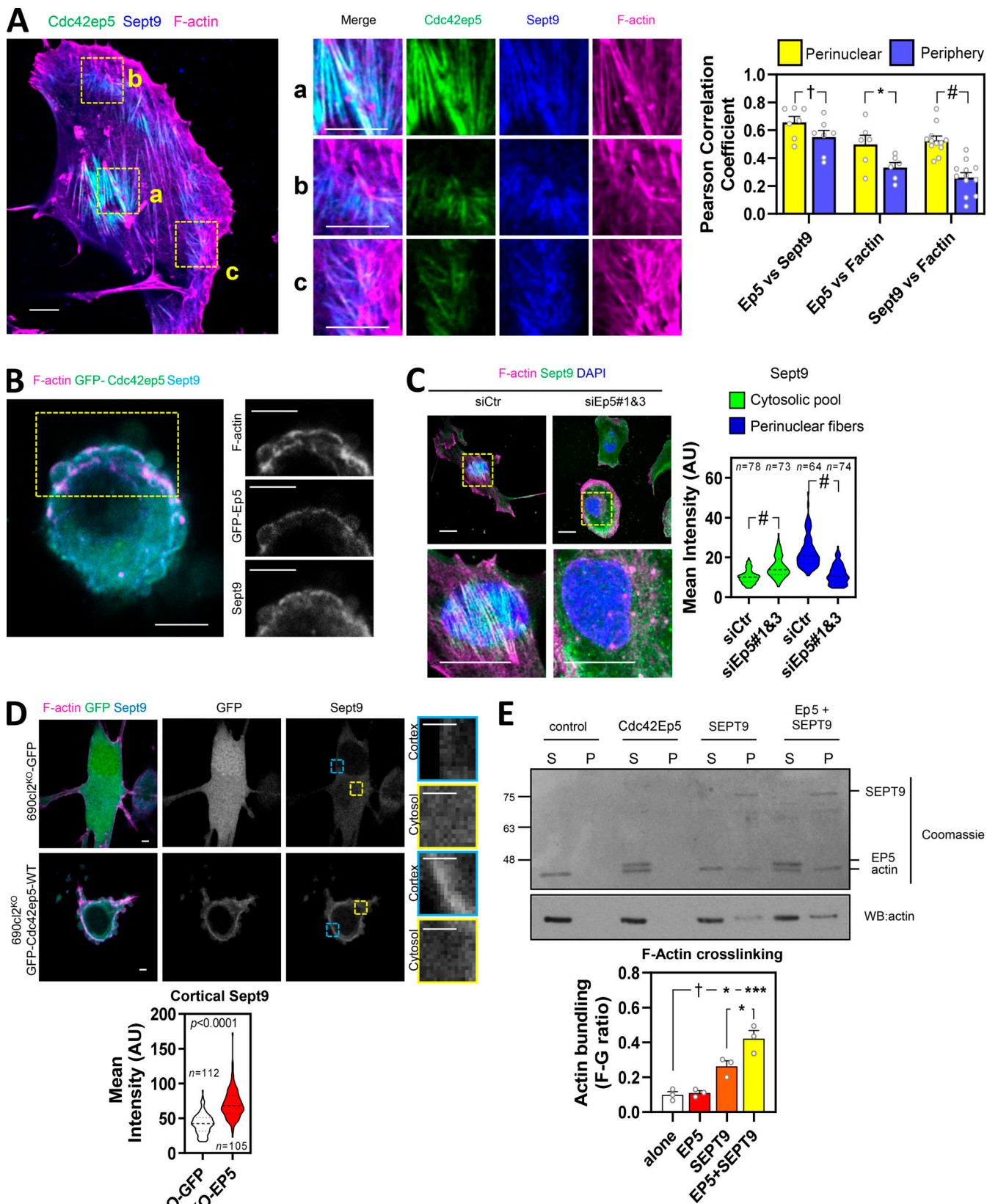

Figure 5.  **CDC42EP5 associates with SEPT9 and F-actin to promote F-actin cross-linking and the generation of actomyosin structures. (A)** Left panel shows a 690.cl2 cell expressing GFP-Cdc42ep5 (green) and stained for Sept9 (blue) and F-actin (magenta). Scale bars, 10 μm. Middle panels show merged and individual channel magnifications of indicated areas: a, perinuclear; b and c, peripheral. Graph shows Pearson's correlation coefficient of Cdc42ep5, Sept9, and F-actin in perinuclear and peripheral areas. Bars indicate mean ± SEM ($n$ = 7, 6, and 12 independent regions in individual cells; unpaired $t$ tests: †, not significant;

*, P < 0.05; #, P < 0.0001. **(B)** Image of 690.cl2 cell expressing GFP-Cdc42ep5 (green) on collagen-rich matrices. Staining of F-actin (magenta) and Sept9 (cyan) and individual channel magnifications of indicated area are also shown. Scale bars, 10 µm. **(C)** Images show F-actin (magenta), Sept9 (green), and DAPI (blue) staining in 690.cl2 cells on glass after transfection with indicated siRNAs. Magnifications of perinuclear areas are shown. Scale bars, 20 µm. Violin plot shows quantifications of cytosolic and perinuclear intensity of Sept9 staining. *n*, individual cells; Mann–Whitney's test: †, not significant; ***, P < 0.001; #, P < 0.0001. **(D)** Images show F-actin (magenta), GFP (green), and Sept9 (cyan) of 690.cl2$^{KO}$ cells expressing GFP or GFP-Cdc42ep5 on collagen-rich matrices. Scale bars, 2.5 µm. Single channels for GFP and Sept9 signals are shown. Panels on the right show magnifications of Sept9 signals in cortical and cytosolic areas. Violin plot at the bottom shows quantification of cortical Sept9 signal. *n*, independent regions in individual cells; Mann–Whitney's test. **(E)** Top panel is a Coomassie-stained gel showing equal volumes of supernatant (S) and pellet (P) fractions from low-speed sedimentation of prepolymerized actin filaments in the presence of recombinant Cdc42ep5 (1 µM), SEPT9 (1 µM), and both Cdc42eep5 (1 µM) and SEPT9 (1 µM). Bottom panel shows an anti-actin Western blot of the same experiment. Graph shows the actin bundling coefficient (ratio of actin in pellet vs. supernatant) in the indicated experimental points. Bars indicate mean ± SEM (*n* = 3 experiments; one-way ANOVA, Tukey's test: †, not significant; *, P < 0.05; ***, P < 0.001). AU, arbitrary units.

actin–septin filament association and enhance MLC2 activation (Joo et al., 2007). In amoeboid lymphocytes, SEPT7 has been shown to modulate the actomyosin cytoskeleton in cortical contraction (Gilden et al., 2012). SEPT9 occupies the terminal position in septin oligomers and has been proposed to be essential for filament formation, although perturbation of SEPT9 function only results in late abscission defects during cytokinesis but does not affect septin-dependent steps earlier in mitosis (Kim et al., 2011). On the contrary, SEPT2, SEPT7, and SEPT11 are required at the early stages of cytokinesis (Estey et al., 2010). Our orthogonal analyses of differential gene expression in clinical samples coupled to loss-of-function characterization indicate that SEPT9 is the main septin effector of CDC42EP5, which was further confirmed using in vitro analyses and epistatic approaches. In the context of melanoma, SEPT9 is essential for the generation of an actomyosin cortex in 3D and stress fibers in 2D. Altogether, it appears that SEPT9 functions in actomyosin networks independently of the formation of canonical SEPT2/6/7/9 oligomers. This specific role appears to be critical in the stabilization of highly contractile actomyosin structures such as the cleavage furrow (Estey et al., 2010), stress fibers (Dolat et al., 2014; Fig. 4), and actomyosin cortex (Fig. 4) and may be related to the unique ability of SEPT9 to cross-link F-actin that is not shared by the rest of septins (Dolat et al., 2014; Smith et al., 2015). In agreement, we demonstrate mechanistically that CDC42EP5 interacts with SEPT9 and potentiates its F-actin bundling activity required for the stabilization of actomyosin networks.

Actomyosin networks such as the actomyosin cortex in 3D and stress fibers in 2D are critical structures for exerting and resisting mechanical tension. In addition to the myosin motors required for generating force, these structures require additional levels of F-actin cross-linking to generate bundles capable of resisting the mechanical loads applied to them. Myosin II is the main actin-based contractile myosin motor that cross-links actin filaments at the cell cortex and stress fibers and regulates cellular tension (Stevenson et al., 2012; Vicente-Manzanares et al., 2009). Accordingly, actomyosin activity in melanoma has primarily been shown to be exerted via Rho-ROCK modulation of myosin activity (Herraiz et al., 2015; Pandya et al., 2017; Sanz-Moreno et al., 2008, 2011; Sanz-Moreno and Marshall, 2010) and is critical in modulating fast amoeboid invasion and metastasis (Sanz-Moreno et al., 2008; Sanz-Moreno and Marshall, 2010). We demonstrate that actomyosin function in melanoma is also dependent on the additional support provided by

SEPT9 in those structures, which is modulated by the septin regulator CDC42EP5. Thus, we observe both CDC42EP5 and SEPT9 colocalizing with actomyosin networks in 2D and 3D, and their silencing is associated with the destabilization of F-actin structures and reduced actomyosin function. Thus, CDC42EP5: SEPT9 filaments appear to complement myosin II function and mechanically stabilize these structures. This new regulatory axis of actomyosin function in melanoma is required for invasion and metastasis.

Septins are increasingly associated with cytoskeletal regulation and cell migration, and potential roles in modulating cancer aggressiveness are emerging (Angelis and Spiliotis, 2016; Spiliotis, 2018). Determining their relevance in different cancers and understanding how they are regulated is therefore critical. We show a specific *SEPT9* up-regulation in human melanoma and particularly in metastatic melanoma, which may underpin its functional relevance in this setting. In addition, we show that melanoma cells present characteristic SEPT9 networks that modulate their migratory behavior by potentiating actomyosin activity. In addition, we identify a new regulatory mechanism of septin function in cancer. Thus, CDC42EP5 is required for the assembly of SEPT9 structures at actomyosin bundles in 2D and 3D, possibly by enabling its correct positioning within the cell or within actin filaments. Thus, silencing of Cdc42ep5 in melanoma cells seeded in 2D leads to the disassembly of higher-order Sept9 filaments, whereas in 3D results in reduced cortical Sept9 (Fig. 7).

Melanoma cells can switch between mesenchymal and amoeboid modes of migration, and this plasticity is a critical factor in the difficulties encountered in the clinic to target metastasis (Gandalovičová et al., 2017). Since actomyosin is required for both migratory modes, our results suggest that the CDC42EP5–SEPT9 axis could potentially reduce both rounded-amoeboid and elongated-mesenchymal modes of invasion. Thus, cells lacking CDC42EP5 or SEPT9 present defects in stress fiber formation and FA maturation that are generally required for mesenchymal migration. In agreement, silencing Cdc42ep5 resulted in a significant decrease in directed migration and motility in 2D, which favors elongated-mesenchymal behaviors. We also describe that CDC42EP5 is required for migration and invasion in a mesenchymal model of breast cancer such as MDA-MB-231-LM2. In breast cancer, SEPT9 has been reported to promote invasion by enhancing ECM degradation through metalloproteinase secretion (Marcus et al., 2019), a hallmark of mesenchymal migratory behaviors. Similar results were

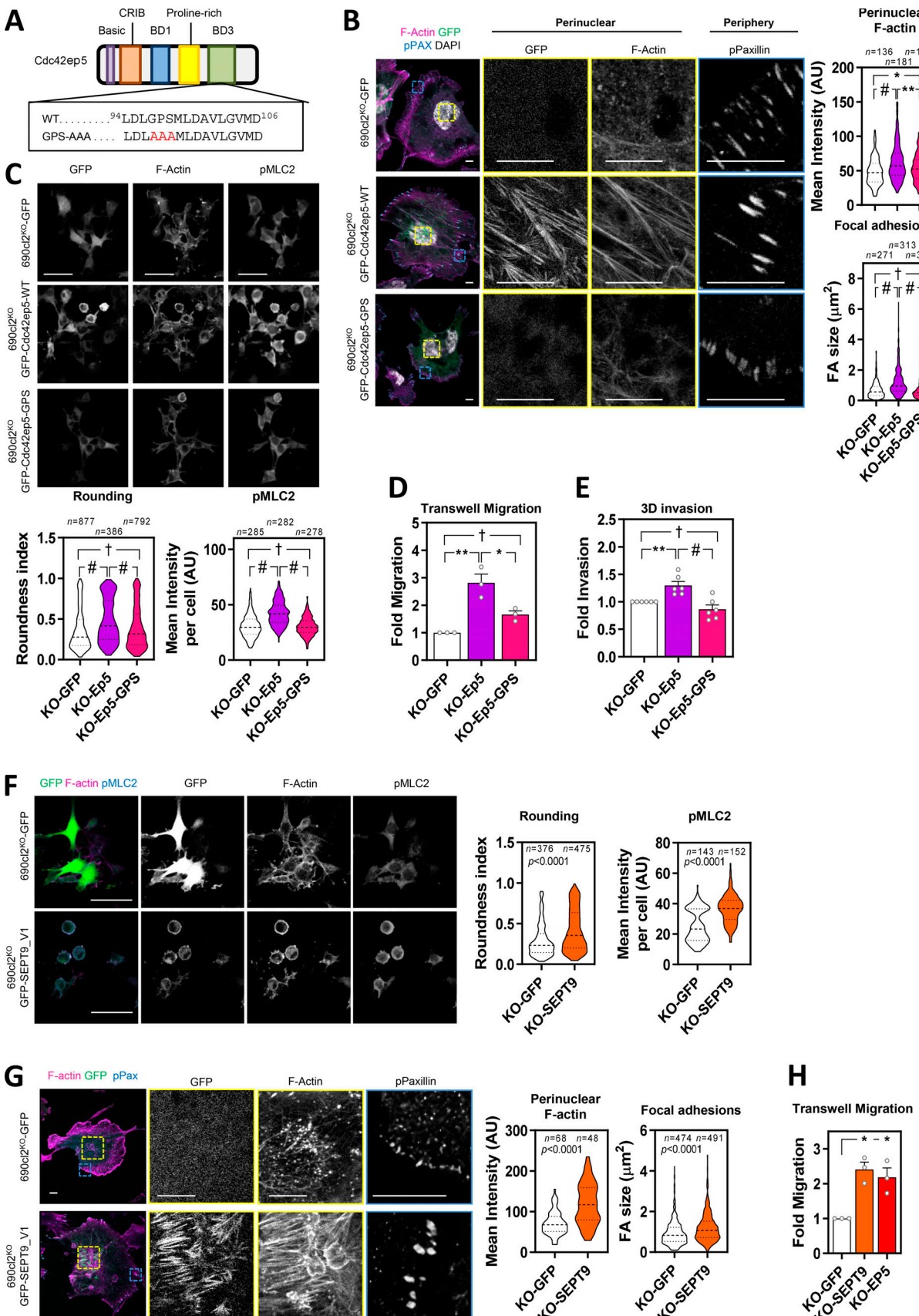

Figure 6. **SEPT9 is a crucial effector of CDC42EP5 function in melanoma. (A)** Diagram showing the different domains in Cdc42ep5. Underneath are the amino acid sequences of the BD3 segment in wild-type and septin-binding–defective mutant (GPS-AAA). **(B)** Images show GFP (green), F-actin (magenta),

pY118-Paxillin (cyan), and DAPI (gray) in 690.cl2KO cells expressing GFP, GFP-Cdc42EP5WT, or GFP-Cdc42EP5GPS-AAA on glass. Single-channel magnifications of indicated areas are shown. Scale bars, 10 µm. Violin plots show quantification of perinuclear F-actin intensity (top) and FA size (bottom). n, independent regions in individual cells; one-way ANOVA Tukey's test: †, not significant; *, P < 0.05; *, P < 0.05; #, P < 0.0001. **(C)** Images show GFP, F-actin and pS19-MLC2 in 690.cl2KO cells expressing GFP, GFP-Cdc42EP5WT, or GFP-Cdc42EP5GPS-AAA on collagen-rich matrices. Scale bar, 75 µm. Bottom graphs show violin plots for roundness index (left) and pS19-MLC2 mean intensity (right) of individual cells. n, individual cells; Kruskal–Wallis and Dunn's tests: †, not significant; #, P < 0.0001. **(D and E)** Graphs show fold migration ability (D) and 3D collagen invasion (E) of 690.cl2KO cells expressing GFP, GFP-Cdc42EP5WT, or GFP-Cdc42EP5GPS-AAA. Bars indicate mean ± SEM (n = 3 [D] and 6 [E]; one-way ANOVA Tukey's test: †, not significant; *, P < 0.05; **, P < 0.01; #, P < 0.0001). **(F)** Images show F-actin (magenta), GFP (green), and pS19-MLC2 (cyan) in 690.cl2KO cells expressing GFP or GFP-SEPT9_V1 on collagen-rich matrices. Individual channels are also shown. Scale bars, 75 µm. Violin plots show roundness index (left) and pS19-MLC2 mean intensity (right) of individual cells. n, individual cells; Mann–Whitney test. **(G)** Images showing GFP (green), F-actin (magenta), and pY118-Paxilin (cyan) in 690.cl2KO cells expressing GFP or GFP-SEPT9_V1 on glass. Right panels show indicated magnifications. Scale bars, 10 µm. Violin plots show quantification of F-actin intensity in the perinuclear region (left) and of FA size (right). n, single regions from individual cells (left) and individual FAs (right); Mann–Whitney test. **(H)** Graph shows fold migration ability of 690.cl2KO cells expressing GFP, GFP-SEPT9_V1, or GFP-Cdc42ep5WT. Bars indicate mean ± SEM (n = 3 experiments; paired t test: *, P < 0.05). AU, arbitrary units. KO, knockout.

obtained in a model of renal epithelial-mesenchymal transition, where SEPT9 was required for mesenchymal migration (Dolat et al., 2014). Together, these findings suggest that the CDC42EP5–SEPT9 axis may be an interesting node to target melanoma metastasis, as it might affect both modes of migration and circumvent the problem of melanoma cell plasticity (Sanz-Moreno and Marshall, 2010).

Although the link between Borg proteins and cytoskeletal rearrangements is longstanding (Joberty et al., 1999), very little is known regarding their relevance in cancer and invasion. Here, we have performed for the first time a comparative study of the ability of different Borg proteins to regulate cancer cell migration. Our analyses identify a unique role for CDC42EP5 in conferring promigratory and proinvasive properties that is not consistently shared by other members of the family. This is surprising, as all Borg proteins share both regulatory and effector domains (Farrugia and Calvo, 2016). We can speculate that specificity in this case is achieved via transcriptional regulation of these genes. Thus, we show a specific up-regulation of *CDC42EP5* in rounded-amoeboid melanoma cells and describe a concomitant function of CDC42EP5 in modulating actomyosin activity in melanoma. This may explain its unique requirement for migration in confined environments such as collagen invasion assays or in vivo migration, as these processes rely significantly in actomyosin function. On the other hand, other Borg proteins may participate in modes of migration that require different cytoskeletal rearrangements. For example, CDC42EP4 function on normal mammary epithelia 2D migration was associated to enhanced filopodia formation (Zhao and Rotenberg, 2014). Still to be determined are the molecular features conferring this unique role to CDC42EP5. This may result from unique binding partners, structural characteristics, or additional regulations that dictate its specific localization and/or function.

## Materials and methods

### Cell lines

Murine melanoma 690.cl2 cells (a kind gift from Richard Marais, Cancer Research UK Manchester Institute, Manchester, UK) were generated from spontaneous melanoma lesions from BrafV600E murine models (Dhomen et al., 2009; Kümper et al., 2016). WM266.4 human melanoma cells (a gift from Chris Bakal, The Institute of Cancer Research, London, UK) harbor a BrafV600D mutation and were isolated from a metastatic lesion (skin) of a female patient who presented with malignant melanoma. MDA-MB-231-LM2 (a gift from Chris Bakal, The Institute of Cancer Research) is a derivative of triple-negative breast cancer cell model MDA-MB-231 that was selected for its ability to metastasize to lung tissue in vivo. Mouse embryonic fibroblasts were a kind gift from Afshan McCarthy (The Institute of Cancer Research, London, UK) and were kept at low passage while avoiding confluency. All of these cell lines were cultured in DMEM (Sigma), GlutaMax (Gibco), and 10% FBS and incubated at 37°C in 5% $CO_2$. A375P and A375M2 melanoma cells were from Richard Hynes (Massachusetts Institute of Technology, Cambridge, MA). CHL, SKMEL28, 501MEL, SKMEL2, SKMEL23, SBCL2, WM1361, WM1366, and WM3670 melanoma cells were

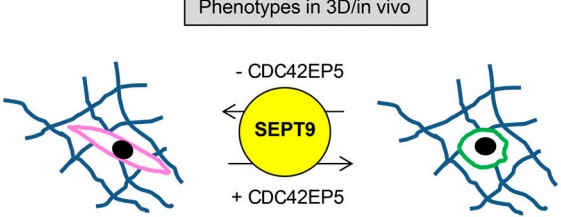

Figure 7. **CDC42EP5 is required for melanoma migration and invasion by promoting actomyosin contractility via SEPT9.** Diagram summarizing the main findings of this study.

from Richard Marais (Cancer Research UK Manchester Institute, Manchester, UK). Cells were maintained in DMEM (RPMI for WM1361, SBCL2, and WM3670) containing 10% FBS and kept in culture for a maximum of three or four passages. All cell lines tested negative for mycoplasma infection with MycoAlert (Lonza).

## cDNA, RNAi, and reagents
Murine pEGFP-Cdc42ep5 (N-terminal GFP) was a kind gift from Facundo Batista and Shweta Aggarwal (Francis Crick Institute, London, UK). This plasmid was used as a template to generate GFP-tagged mutant versions Cdc42ep5[GPS-AAA] using In-Fusion cloning (Takara). To allow for lentiviral infection and stable expression, these cDNAs were all then subcloned into a pCSII-IRES-blasti vector backbone at NheI and BamHI restriction sites. The wild-type version of Cdc42ep5 was also subcloned in-frame into pGEX backbone (pGEX-GST-Cdc42ep5) for generation of recombinant protein. Details of cloning strategy are available upon request. GFP-SEPT9_V1 plasmid was a kind gift from Cristina Montagna (Albert Einstein College of Medicine, Bronx, NY). Paxillin-GFP was a kind gift from Chris Bakal (The Institute of Cancer Research, London, UK). pCSII-IRES2-blasti-GFP and pCSII-IRES2-blasti-mCherry were a kind gift from Erik Sahai (Francis Crick Institute, London, UK). pnCS-Strep-SEPT6/7 is a bicistronic, spectinomycin-resistant plasmid that expresses human SEPT6 and strep-SEPT7 in tandem; pET28-SEPT9 encodes a kanamycin-resistant His-tagged version of human SEPT9 (Mavrakis et al., 2016). Both plasmids were a kind gift from Elias Spiliotis (Drexel University, Philadelphia, PA). siRNAs were purchased from Dharmacon and are listed in Table S1.

## Generation of CRISPR knockout cell lines
The CRISPR plasmid U6gRNA-Cas9-2A-GFP containing a guide RNA targeting murine Cdc42ep5 was purchased from Sigma (MM0000377239). Parental 690.cl2 cells were transfected with that plasmid using Lipofectamine (Life Technologies) following the manufacturer's instructions, and GFP-positive cells were single sorted into 96-well plates after 24 h. Individual cell clones were expanded and the Cdc42ep5 locus targeted by CRISPR was sequenced and subjected to Western blot analysis for knockout validation (690.cl2[KO]). Cdc42ep5-null clones were infected with GFP-Cdc42ep5–expressing lentivirus to generate 690.cl2[KO] cells reexpressing wild-type Cdc42ep5 (690.cl2[KO]-GFP-Cdc42ep5) or the septin-binding–defective mutant of Cdc42ep5 (690.cl2[KO]-GFP-Cdc42ep5[GPS-AAA]). Alternatively 690.cl2[KO] cells were infected with plain GFP-expressing lentivirus to generate 690.cl2[KO]-GFP cells.

## Transfections
Cells were seeded at 75% confluency and transfected using RnaiMax (Life Technologies) for siRNA (100 nM final concentration) and Lipofectamine 3000 (Life Technologies) for plasmids following the manufacturer's instructions. Cell lines stably expressing cDNA (GFP-Cdc42ep5 or GFP/mCherry) were generated by lentiviral infection followed by blasticidin selection for 2 wk (4 µg ml$^{-1}$). Alternatively, fluorescent-labeled cells were

sorted by FACS. For the generation of the nonvirally transduced stable 690.cl2 cells expressing GFP, GFP-Paxillin, or GFP-SEPT9_V1, parental cells were transfected with the relevant plasmids using Lipofectamine 3000 following the manufacturer's instructions. Next, 48 h after transfection, 400 µg ml$^{-1}$ G418 (Sigma) was added as a selection agent to remove non-transfected cells. Cells were probed for the expression of plasmids based on GFP expression and kept at low passage.

## Proliferation assay
Cells were seeded in a 96-well in triplicate at 5,000 cells per well and grown in normal culture conditions. AlamarBlue (Thermo-Fisher) assay was used according to manufacturer's instructions to compare cell growth and viability at the indicated time points. Samples were run in quadruplicate and averaged. Values were normalized to values at day 0.

## Transwell migration
$5 \times 10^4$ cells were suspended in serum-free media and seeded onto transwell inserts with 8-µm pores (Corning) on a reservoir containing 10% FBS and 10 ng ml$^{-1}$ TFGβ (Peprotech). After 48 h, the cells on the lower chamber were fixed in 4% PFA for 15 min, stained with DAPI and imaged using a confocal multiphoton inverted microscope with a motorized XY scanning stage and four nondescanned detectors (two Hybrid Detectors [HyD] + two photomultiplier tubes [PMT]; Leica TCS SP8). A tile scan containing the entire well was acquired with a 10×/0.4 NA objective lens using LAS X software (Leica). The number of cells (i.e., nuclei) was quantified using ImageJ. Data are expressed as mean ± SEM (normalized to control cells).

## 3D inverted invasion assay
For 690.cl2, cells were suspended in 2.3 mg ml$^{-1}$ rat tail collagen at $10^5$ cells ml$^{-1}$. 100-µl aliquots were dispensed into 96-well flat bottom plates (Corning) precoated with 0.2% fatty acid–free BSA (Sigma). Cells were spun to the bottom of the well by centrifugation at 300 ×$g$ for 5 min at 4°C and incubated at 37°C in 5% CO$_2$ for 1 h. Once collagen had polymerized, DMEM with 10% FBS and TGF-β (10 ng ml$^{-1}$) was added on top of the collagen. After 24-h incubation at 37°C in 5% CO$_2$, cells were fixed and stained for 4 h in 16% PFA solution containing 5 µg ml$^{-1}$ Hoechst 33258 nuclear stain (Invitrogen). The plates were then imaged using an Operetta High Content Imaging System (HH12000000; Perkin-Elmer) and Harmony software (PerkinElmer) at z-planes 0, 30, and 60 µm. The number of cells at each plane (i.e., nuclei) was quantified using the Harmony software package (PerkinElmer). The invasion index was calculated as the sum of the number of cells at 60 µm divided by the total number of cells (at 0, 30, and 60 µm). The invasion index of each condition was then normalized to the respective control. Samples were run in triplicate and averaged. Data are expressed as mean ± SEM (normalized to control cells). Alternatively, cells were stained with phalloidin-tetramethylrhodamine (TRITC) to generate 3D reconstructions of collagen invasion assays. Z-stacks (1 µm size) were acquired from bottom to top using a Leica TCS SP8 confocal microscope with 20x/0.75 NA objective lens, and 3D reconstructions generated using LAS X software (Leica).

For WM266.4 and MDA-MB-231-LM2 cells: 100 µl of diluted 1:1 (Matrigel/PBS) mix was pipetted into transwell inserts in a 24-well tissue culture plate and left to set for 30 min at 37°C. When the Matrigel had set, the transwell inserts were inverted and $5 \times 10^4$ cells in normal media were pipetted onto the filter. The transwell inserts were then carefully covered with the base of the 24-well tissue culture plate, making contact with each cell suspension droplet, and the plate incubated in the inverted state for 4 h to allow cell attachment. After this, plates were turned upside up, and each transwell insert was washed three times with 1 ml serum-free DMEM and finally placed in a well containing serum-free DMEM. Full normal media (10% FBS) was gently pipetted on top of the set Matrigel/PBS mixture and incubated for 5 d at 37°C in 5% $CO_2$. After this, transwell inserts were fixed in 4% PFA, stained with DAPI and phalloidin-TRITC, and imaged by confocal microscopy (TCS SP8; Leica) with 20×/0.75 NA objective lens. Optical Z-sections were scanned at 15-µm intervals moving up from the underside of the filter into the Matrigel. The invasion index was calculated as the sum of the area of cells at 30 µm divided by the total number of cells (at 0, 15, and 30 µm). Data are expressed as mean ± SEM (normalized to control cells).

### ECM-remodeling assay
To assess force-mediated matrix remodeling, $3 \times 10^5$ cells were embedded in 120 µl of a 2.3 mg ml$^{-1}$ rat tail collagen-1 gel in 24-well glass-bottom MatTek dishes and incubated at 37°C in 5% $CO_2$ for 1 h. Once the gel was set, cells were maintained in normal culture conditions. Gel contraction was monitored daily by imaging the plates with a desktop scanner. The gel contraction value refers to the contraction observed after 2 d. To obtain the gel contraction value, the relative diameters of the well and gel were measured using ImageJ software, and the percentage of contraction was calculated using the formula [100 × (well area – gel area)/well area]. Data are expressed as mean ± SEM.

### Generation of xenograft tumors and intravital imaging
All animals were kept in accordance with UK regulations under project license PPL80/2368. 6–8-wk-old CD1 nude mice were injected subcutaneously with $10^6$ 690.cl2$^{KO}$-GFP or 690.cl2$^{KO}$-Cdc42ep5 murine melanoma cells suspended in 100 µl PBS/Matrigel (50:50). Tumor size was measured every other day using calipers. To calculate tumor volume, the formula [V = (length × width$^2$)/2] was used. Intravital imaging was performed using a confocal multiphoton inverted microscope with a motorized XY scanning stage, four-channel spectral scanhead (descanned), and two hybrid detectors (TCS SP8; Leica) when tumors reached 6–8 mm. Tumors were excited with an 880-nm pulsed Ti–Sapphire laser and emitted light acquired at 440 nm (collagen second harmonic generation) and 530 nm (GFP) using 20×/0.75 NA objective lens. During ~10-min intervals, five to eight different regions were imaged simultaneously for 2 h for each tumor. In each region, a z-stack of three images (~50 µm deep on average) was taken, resulting in a time-lapse 3D z series for analysis. Time-lapse movies were processed and analyzed using ImageJ, including a "3D drift correction" script. This was achieved by converting the images obtained to hyperstacks and correcting for drift using the static second harmonic generation signal. The images generated were then processed to generate movies using LAS X software. To generate colored time projections, time-lapse movies were loaded into ImageJ and the function "Hyperstack>Temporal-Color Code" used. This generates motion analysis images by overlaying blue, green, and red images from different time points. Distinct areas of color indicate motile cells, whereas white areas indicate static regions. Once processed, images were assessed for cell movement, cell morphology, and cell size. Moving cells were defined as those that moved ≥10 µm during the length of each movie, and the moving distance and resulting speed was determined using LAS X software.

### Experimental metastasis assay
690.cl2 cells stably expressing GFP or mCherry were transfected with control or experimental siRNA (Cdc42ep5 or Sept9), respectively. 48 h after transfection, Cherry and GFP cells were mixed in PBS at a ratio of 1:1, and $10^6$ cells (mixed population) were injected into the tail vein of CD1 nude mice. Mice were culled 2 and 24 h after injection, and lungs were dissected and placed in PBS. GFP and Cherry signal in fresh lungs were collected using a Leica TCS SP8 confocal inverted microscope (20×/0.75 NA objective lens; 12 fields of view per lung). The area of GFP and Cherry cells was quantified using Volocity. For each mouse, the percentage of GFP and Cherry area of both lungs was averaged. Data are expressed as mean ± SEM from at least four independent animals.

### Time-lapse analysis and cell migration in 2D
All live-imaging experiments were performed at 37°C 5% $CO_2$ in normal growth medium. Time-lapse microscopy was performed with an inverted microscope (TE 2000; Nikon) equipped with a motorized stage, a camera (OrcaR2; Hamamatsu Photonics) and HCImaging software. Cells were seeded at low density, and bright-field images were captured every hour for 18 h with 20×/0.45 NA objective lens. Individual cells were tracked using the MTrackJ ImageJ plugin.

### RNA isolation and quantitative RT-PCR
RNA was isolated using RNeasy Kit (Qiagen). Reverse transcription was performed using Precision NanoScript 2 Reverse-Transcription kit (PrimerDesign) and quantitative PCR using the PrecisionPLUS 2x qPCR MasterMix with ROX and SybrGreen (PrimerDesign). Expression levels of indicated genes were normalized to the expression of *Gapdh/GAPDH* or *Rplp1*. Sequences of the oligonucleotides used for quantitative RT-PCR are described in Table S2. Data on Fig. S1 I describing the RSEM (RNA-seq by expectation maximization) values from melanoma and breast cancer samples were retrieved from The Cancer Genome Atlas.

### Coimmunoprecipitation
690.cl2 cells expressing the GFP-tagged plasmids of interest were grown on 150-mm Petri dishes and lysed in lysis buffer (50 mM Tris-HCl, pH 7.5, 150 mM NaCl, 1% [vol/vol] Triton X-100, 10% [vol/vol] glycerol, 2 mM EDTA, 25 mM NaF, and 2 mM

NaH$_2$PO$_4$). The resultant lysates were first precleared using IgG-conjugated Protein G beads, incubated with the specific antibodies for 2 h at 4°C, and then incubated with Protein G beads for 1 h at 4°C with gentle mixing. Beads were then washed four times with lysis buffer and eluted with 20 µl of 2X SDS sample buffer.

### Generation of recombinant proteins
*Escherichia coli* BL21(DE3) bacteria transformed with pGEX-GST-Cdc42ep5, pnCS-Strep-SEPT6/7, or pET-28a-His-SEPT9 were grown in LB medium at 37°C with ampicillin, spectinomycin, and kanamycin, respectively. Protein expression was induced at an OD$_{600}$ of 0.8 with 0.5 mM IPTG for 16 h at 18°C, harvested by centrifugation, and suspended in 50 mM Tris-HCl, pH 8, 300 mM NaCl, 1 mM PMSF, and 5 mM benzamidine. Bacteria were sonicated four times on ice (amplitude 20%, 15 s on, 45 s off) and centrifuged at 20,000 ×g for 30 min at 4°C. Supernatants were collected and incubated with 400 µl of the respective beads (Glutathione Sepharose, Strep-Tactin, or NiNTA agarose) for 1 h. Beads were washed five times with two column volumes of lysis buffer and eluted with two column volumes of 50 mM Tris-HCl, pH 8, 300 mM KCl, and 5 mM MgCl$_2$ supplemented with imidazole (250 mM), glutathione (10 mM), or desthiobiotin (2.5 mM). The eluted proteins were further purified by size-exclusion chromatography on a Superdex 200 Increase 10/300 GL column (GE Healthcare) in 50 mM Tris-HCl, pH 8, 300 mM KCl, 5 mM MgCl$_2$, and 5 mM DTT. The concentration of all purified proteins was determined by the Bradford assay.

### In vitro actin assays
Human non-muscle actin (Cytoskeleton) in general actin buffer (5 mM Tris-HCl, pH 8.0, and 0.2 mM CaCl$_2$ supplemented with 0.2 mM ATP) was polymerized using actin polymerization buffer (50 mM KCl, 2 mM MgCl$_2$, and 1 mM ATP) at 24°C for 1 h to generate F-actin. This stock of F-actin was used for the bundling and binding assay. For the bundling assay, indicated concentrations of the different recombinant proteins (GST-Cdc42ep5, His-SEPT9, or Strep-SEPT6/7) were incubated with 2 µM F-actin for 1 h at 24°C. Samples were centrifuged at 14,000 ×g for 1 h at 24°C. For the F-actin–binding assay, the reaction of actin and the different recombinant proteins was centrifuged at 150,000 ×g for 1.5 h at 24°C. For the actin polymerization assay, actin was incubated on ice for 1 h with general actin buffer to generate the G-actin stock. This stock was then mixed with the actin polymerization buffer and the indicated concentrations of the recombinant proteins at 24°C for 30 min. Samples were centrifuged at 150,000 ×g for 1.5 h at 24°C. In all cases, equal volumes of the supernatant and pellet fractions were resolved using a 12% SDS-PAGE gel, stained with Coomassie brilliant blue, and processed by Western blotting (for GST, His, and actin) if indicated.

### Western blotting
Protein lysates and immunoprecipitants were run on SDS-PAGE gels (10% or 12%), transferred to nitrocellulose membranes, and probed for the indicated proteins. Enhanced chemiluminescence signal was acquired using an Azure Biosystems c600. Exposures within the dynamic range were quantified by densitometry using ImageJ. Antibody descriptions and working dilutions can be found in Table S3.

### Immunofluorescence
Cells were seeded on glass-bottom 24-well plates (MatTek). Where indicated, cells were seeded in glass-bottom plates covered with 10 µg ml$^{-1}$ fibronectin. For analysis of 3D morphology, cells were seeded on top of 2.3 mg ml$^{-1}$ collagen-I gel over a glass-bottom dish (MatTek) in medium and allowed to adhere for 24 h. Cells were fixed in 4% PFA and permeabilized in PBS with 0.2% Triton X-100. Where indicated (i.e., septin staining) cells were fixed in ice-cold methanol for 15 min and permeabilized in PBS with 0.2% Triton X-100. Samples were blocked in 3% BSA with 0.1% PBS Tween (PBST) for 3 h. Primary antibodies (Table S3) were diluted in 3% BSA in PBST for 2 h. Wells were then washed three times for 10 min each in 3% BSA PBST, followed by the addition of the appropriate secondary (Alexa Fluor; Invitrogen). After three 15-min washes in PBS, samples were mounted and analyzed.

### Microscopy and image analysis
For morphological analyses of cells grown over collagen, a minimum of five images were taken of cells per experiment using 10×/0.40 NA or 20×/0.75 NA objective lenses on an inverted confocal microscope (TCS SP8; Leica). For analyses of fluorescence intensity differences, microscope settings were kept constant and independent replicates imaged on the same day. Using phalloidin-TRITC staining and ImageJ software, individual cells were selected and the roundness coefficient obtained using the circularity function [4pi (area/perimeter$^2$)]. For analysis of single-cell F-actin/pS19-MLC2/GFP intensity, individual cells were selected using ImageJ software and the mean fluorescence intensity of each channel per cell. Volocity was used for analysis of SEPT9 cytosolic/cortical intensity; areas in the center of the cell (excluding the nucleus) were determined for the cytosolic region, and small regions toward the cell periphery (based on F-actin staining) were chosen to determine the cortical region of the cell. Cytosolic/cortical areas between samples were kept constant. The mean fluorescence intensity of Sept9 within individual areas was determined by the software.

For analysis of actin/septin/Cdc42ep5 structures in cells on 2D, cells were stained for F-actin/Sept9 and imaged at the basal plane using 63×/1.40 NA oil immersion objective lens (TCS SP8; Leica). The basal perinuclear region of individual cells (based on the DAPI signal) was identified in each cell as the perinuclear region; the peripheral area was determined as a region close to the cell border and not in the vicinity of the nucleus. The fluorescence intensity of individual channels was determined using Volocity in the different areas. For analyses of fluorescence intensity differences, microscope settings were kept constant and independent replicates imaged on the same day.

For analysis of FA size, cells were stained with pY118-Paxillin antibody, imaged using a 63×/1.40 NA oil immersion objective lens (TCS SP8 microscope; Leica), and analyzed using Volocity to determine individual FA area. To determine the maturity of FAs,

Zyxin staining was used. A minimum of five fields of view per repetition were imaged using 63×/1.40 NA oil immersion objective lens (TCS SP8 microscope; Leica), and cells were scored based on the presence or absence of Zyxin adhesions.

To determine the dynamics of Cdc42ep5 and F-actin, parental 690.cl2 cells were cotransfected with GFP-Cdc42ep5 and MARS-LifeAct. Experiments were performed at 37°C 5% $CO_2$ in normal growth medium. Cells were imaged with total internal reflection fluorescence (TIRF) microscopy with Zeiss Axio Observer Z1 microscope equipped with 63×/1.46 NA oil immersion objective lens, motorized stage, Hamamatsu Flash 4 v3 cameras, and Slidebook 6.0 software (3i Imaging Solutions). Fluorescent images were acquired every second for 60 s. Videos were rendered using ImageJ software.

To determine FA dynamics, a stable 690.cl2-GFP-Paxillin cell line was used. Cells were transfected with either control or RNAi targeting Cdc42ep5. TIRF was used to visualize adhesion dynamics using the same equipment as before. Cells were imaged for 10 min (one frame every 30 s). Videos were rendered using ImageJ software. To generate colored time projections, time-lapse movies were loaded into ImageJ and the function "Hyperstack>Temporal-Color Code" used.

For colocalization analyses, cells were stained and imaged as before and processed using ImageJ software using the "Coloc2 plugin" (Pearson's R value no threshold). For each individual cell, three independent regions of interest (perinuclear, peripheral, or encompassing a FA) were analyzed and the average value used as the Pearson's coefficient in that cell. Image acquisition and postprocessing settings and region of interest size were maintained constant for each condition.

### Analysis of clinical datasets of melanoma

Gene expression data from previous studies (GSE3189, Talantov et al., 2005; GSE7553, Riker et al., 2008) were retrieved from NCBI GEO. Gene expression values for septin genes were calculated as a combination of all probes available for each gene. Briefly, all probes capturing septin gene expression on individual samples were z-score normalized. Final values of expression for individual septin genes in each sample were calculated by summing the z-scores of all probes available for each gene. *CDC42EP5* and some septin genes (i.e., *SEPT1*, *SEPT3*, *SEPT12*, and *SEPT14* for Talantov et al. and *SEPT14* for Riker et al.) were not probed in these studies and therefore were not included in the analyses.

### Statistical analysis

Statistical analyses were performed using GraphPad Prism (GraphPad Software). When *n* permitted, values were tested for Gaussian distribution using the D'Agostino–Pearson normality test. When normality tests were not possible, data distribution was assumed to be normal. For Gaussian distributions, paired or unpaired two-tailed Student's *t* test and one-way ANOVA with Tukey posttest (for multiple comparisons) were performed. Following the software recommendations, Geisser–Grenhouse correction and Dunnett's multiple comparisons test were applied in paired one-way ANOVA. For non-Gaussian distributions, Mann–Whitney and Kruskal–Wallis tests with Dunn's posttest

(for multiple comparisons) were performed. Unless stated otherwise, mean values and SEM are shown. Violin plots were generated using GraphPad; in addition to showing the distribution, thick dashed lines represent the median, whereas thin dashed lines represent the upper and lower quartiles. P values < 0.05 were considered statistically significant (*, $P < 0.05$; **, $P < 0.01$; ***, $P < 0.001$; #, $P < 0.0001$; †, nonsignificant). In each graph, the number of individual experimental points is described.

### Data and materials availability

The datasets and materials generated and/or analyzed during the current study are available from the corresponding author upon reasonable request. Materials may be subjected to Material Transfer Agreements.

### Online supplemental material

Fig. S1 shows additional information on the role of CDC42EP5 in melanoma migration and invasion. Fig. S2 shows the effect of CDC42EP5 on actomyosin function in cells seeded on collagen-rich matrices. Fig. S3 provides additional information on the modulation of cytoskeletal features by CDC42EP5. Fig. S4 shows data on septins and melanoma aggressiveness. Fig. S5 shows in vitro analysis of actin binding, polymerization, and bundling by SEPT6/7, SEPT9, and Cd42ep5. Fig. S6 shows how CDC42EP5 interacts with SEPT9 to promote actomyosin activity. Video 1 shows intravital imaging of 690.cl2 cells expressing Cdc42ep5 in a living tumor. Video 2 shows intravital imaging of 690.cl2 cells lacking Cdc42ep5 in a living tumor. Video 3 shows dynamics of LIFEACT-RFP and GFP-Cdc42ep5 in 690.cl2 cells. Video 4 shows GFP-Paxillin dynamics in control and Cdc42ep5-depleted 690.cl2 cells. Table S1 describes RNAi, Table S2 describes oligonucleotides, and Table S3 describes antibodies used in this study.

## Acknowledgments

We thank Facundo Bastista, Elias Spiliotis, Erik Sahai, Cristina Montagna, and Chris Bakal for providing us with plasmids; Chris Bakal, Afshan McCarthy, Richard Hynes, and Richard Marais for providing us with cell lines; Fredrik Wallberg (Light Microscopy Unit at The Institute of Cancer Research) and Victor Campa (Microscopy Unit at Instituto de Biomedicina y Biotecnología de Cantabria) for assistance; and members of the Biological Services Unit at The Institute of Cancer Research for help with mouse experiments. We also thank present and past laboratory members and Chris Bakal for help and advice throughout this work.

This work was funded by the Institute of Cancer Research (A.J. Farrugia, J. Rodríguez, and F. Calvo). F. Calvo is also funded by the Ramon y Cajal Research Program (RYC-2016-20352; Formación Sanitaria Especializada, Agencia Estatal de Investigación Agencia Estatal de Investigación), Fondo Europeo de Desarrollo Regional (RTI2018-096778-A-I00), and Cancer Research UK (C57744/A22057). J.L. Orgaz and V. Sanz-Moreno were supported by grants from Cancer Research UK (C33043/A12065 and C33043/A24478), the Harry J. Lloyd Charitable Trust, and Barts Charity. M. Lucas was supported by Formación

Sanitaria Especializada, Agencia Estatal de Investigación, (RYC-2016-20342) and Ministerio de Ciencia, Innovación y Universidades, Agencia Estatal de Investigación, Fondo Europeo de Desarrollo Regional (RTI2018-097801-B-I00).

The authors declare no competing financial interests.

F. Calvo conceived the study. A.J. Farrugia and F. Calvo designed, performed, and analyzed the experiments, with help from J.L. Orgaz and V. Sanz-Moreno. J. Rodríguez generated recombinant proteins and performed in vitro analyses with help from M. Lucas. F. Calvo wrote the manuscript with essential contribution by A.J. Farrugia and V. Sanz-Moreno.

Submitted: 28 December 2019

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

# Supplemental material

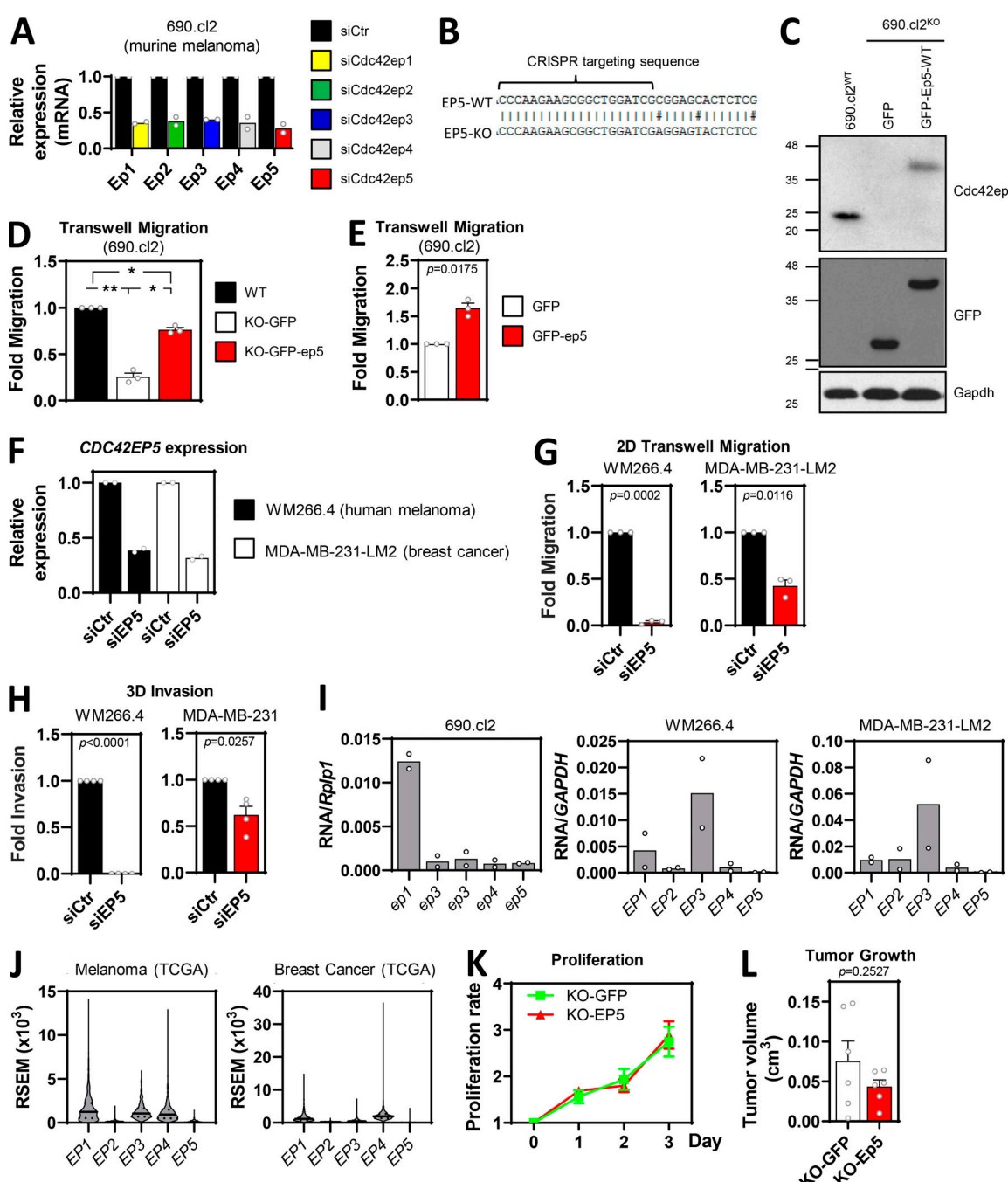

Figure S1. **CDC42EP5 is required for melanoma migration and invasion. (A)** Efficacy of RNAi silencing of Borg genes in 690.cl2 cells. Graph shows fold normalized expression of *Cdc42ep1-5 (Ep1-5)* against control cells (siCtr) cells when individual genes were targeted (siEp1-5). Bars indicate mean (*n* = 2 independent experiments). **(B)** Diagram showing the targeting sequence for endogenous Cdc42ep5 CRISPR/CAS9 knockout in murine wild-type 690.cl2 cells. Underneath is the sequence of the same locus in the 690.cl2$^{KO}$ cells. **(C)** Western blot showing Cdc42ep5, GFP, and Gapdh expression in parental 690.cl2 cells (wild-type) and in 690.cl2$^{KO}$ cells expressing GFP or GFP-Cdc42ep5. **(D)** Graph shows fold migration ability of wild-type 690.cl2 (WT) cells compared with 690.cl2$^{KO}$ cells expressing GFP or GFP-Cdc42ep5. Bars indicate mean ± SEM (*n* = 3 experiments; one-way paired ANOVA Tukey's test; *, P < 0.05; **, P < 0.01). **(E)** Graph shows migration ability of parental 690.cl2 cells ectopically expressing GFP or GFP-*CDC42EP5*. Bars indicate mean ± SEM (*n* = 3 experiments; paired *t* test). **(F)** Graph shows fold normalized expression of Cdc42EP5 in human WM266.4 and MDA-MB-231-LM2 cells after transfection with control (siCtr) and CDC42EP5 (siEP5) siRNAs. Bars indicate mean (*n* = 2 independent experiments). **(G and H)** Graphs show fold migration ability (G) and 3D invasion (H) of WM266.4 and MDA-MB-231 cells after transfection with control (siCtr) and CDC42EP5 (siEP5) siRNAs. Bars indicate mean ± SEM (*n* = 3 experiments [G] and 4 [H]; paired *t* tests). **(I and J)** Graphs show expression of Borg genes (*CDC42EP1-5, EP1-5*) in the indicated cell lines (I) and databases (J). Expression levels in 690.cl2, WM266.4, and MDA-MB-231-LM2 were normalized against expression of *Rplp1* and *GAPDH*, as indicated. Expression levels in melanoma and breast cancer were obtained from The Cancer Genome Atlas and expressed in RSEM (RNA-seq by expectation maximization). **(K)** Cell proliferation curves of 690.cl2$^{KO}$ cells ectopically expressing GFP (KO) or GFP-Cdc42ep5 (KO-EP5). Lines indicate mean ± SEM (*n* = 3 experiments). **(L)** Graph shows quantification of volumes at day 7 post-injection of subcutaneous tumors induced by injection of 690.cl2$^{KO}$ cells expressing GFP or GFP-Cdc42ep5. Bars indicate mean ± SEM (*n* = 6 mice; paired *t* test). KO, knockout.

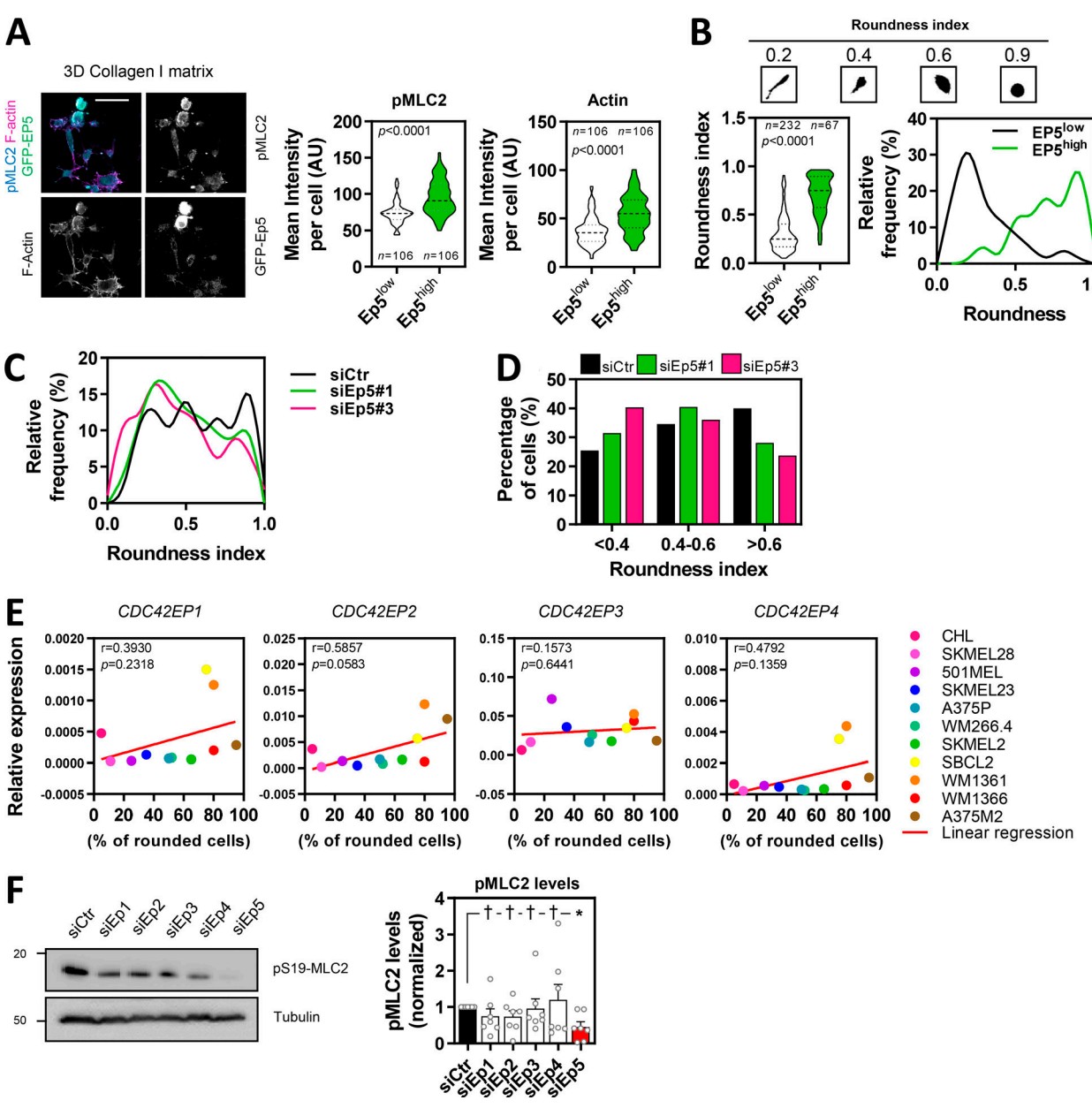

Figure S2. **CDC42EP5 promotes actomyosin function in collagen-rich matrices. (A)** Images of 690.cl2 cells expressing GFP-Cdc42ep5 (green) on collagen-rich matrices. Merged and single greyscale channels also show F-actin (magenta) and pS19-MLC2 (cyan) staining. Scale bar, 50 µm. Violin plots show pS19-MLC2 (left) and F-actin (right) mean intensity from individual cells in 690.cl2 cells expressing different levels of Cdc42ep5 (high or low). *n*, individual cells; Mann–Whitney test. **(B)** Top panels represent the different shape of individual 690.cl2 cells and their respective roundness index. Left graph shows a violin plot of the roundness index of individual cells in 690.cl2 cells expressing different levels of Cdc42ep5 (high or low). *n*, individual cells; Mann–Whitney test. Right graph shows the relative frequencies of roundness indexes in Cdc42ep5hi and Cdc42ep5lo 690.cl2 cells. **(C)** Graph shows the relative frequencies of roundness indexes in 690.cl2 cells after transfection with control (siCtr) or two individual Cdc42ep5 (siEp5) siRNAs (additional representation of Fig. 2 C). **(D)** Graph shows the percentage of cells within three different ranges of roundness, from elongated (<0.4) to rounded (>0.6), in 690.cl2 cells after transfection with control (siCtr) or two individual Cdc42ep5 (siEp5) siRNAs. Bars represent mean. Additional representation of Figs. 2 C and S2 D. **(E)** Graphs show expression of *CDC42EP1*, *CDC42EP2*, *CDC42EP3*, and *CDC42EP4* normalized to *GAPDH* expression in human melanoma cell lines with increasing rounding coefficients. Pearson's correlation coefficient (*r*), statistical significance (*p*), and linear regression (red line) are shown. Each point in the graph represents the mean value of three independent experiments. **(F)** Western blot of indicated proteins in 690.cl2 cells after transfection with control (siCtr) and siRNAs targeting individual Borg genes (siEp1–5). Graph shows the quantification of normalized pS19-MLC2 levels in the different experimental points. Bars indicate mean ± SEM (*n* = 7 experiments; one-way paired ANOVA, Dunnett's test: †, not significant; *, P < 0.05). AU, arbitrary units.

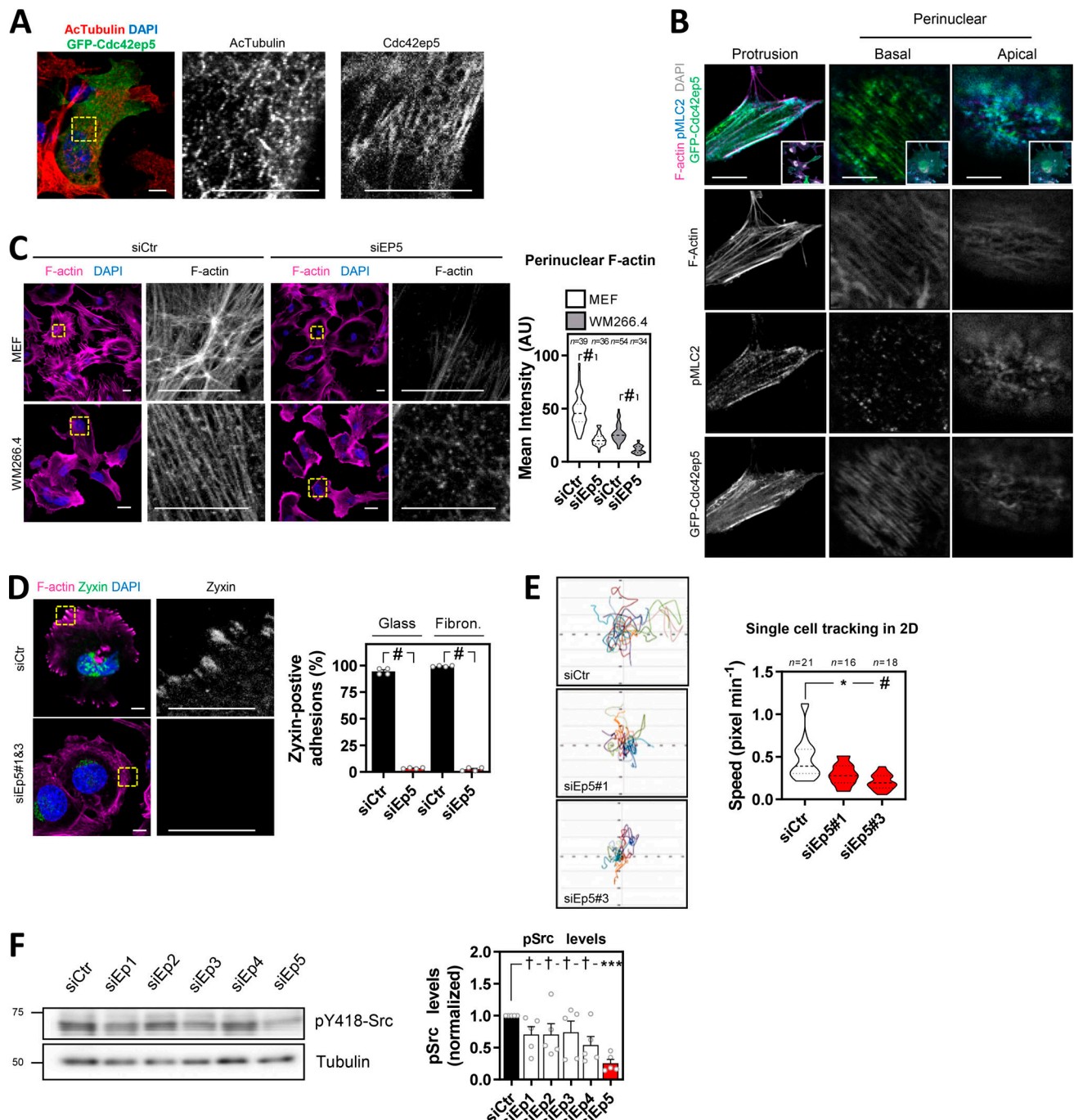

Figure S3. **CDC42EP5 modulates cytoskeletal features. (A)** Image showing acetylated tubulin (red), GFP (green) and DAPI (blue) in 690.cl2 expressing GFP-Cdc42ep5 on glass. Right panels show single greyscale channel magnifications of the indicated area. Scale bars, 12.5 µm. **(B)** Images show F-actin (magenta), GFP (green), pS19-MLC2 (cyan), and DAPI (gray) signal of 690.cl2 cells stably expressing GFP-Cdc42ep5 on glass. Top panels show merged channel images of a protrusion (left) and perinuclear areas (basal, middle panel; apical, right panel). Bottom right inserts show whole-cell images. Lower panels show individual greyscale channels for the indicated regions and signals. Scale bars, 10 µm. **(C)** Images show F-actin (magenta) and DAPI (blue) staining of mouse embryonic fibroblasts (MEFs; top panels) and human melanoma cells (WM266.4, lower panels) on glass after transfection with control (siCtr) and CDC42EP5 (siEP5) siRNA. Grayscale magnifications of F-actin staining in indicated perinuclear areas are shown. Scale bars, 12.5 µm. Violin plot shows perinuclear F-actin intensity. $n$, single regions from individual cells; Mann–Whitney test: #, $P < 0.0001$. **(D)** Images of 690.cl2 cells after transfection with control (siCtr) and Cdc42ep5 (siEp5#1&3) siRNA on glass. Images show F-actin (magenta), Zyxin (green), and DAPI (blue) staining. Right panels show greyscale Zyxin channel magnifications. Scale bars, 12.5 µm. Graph shows percentage of cells with Zyxin-positive FAs for cells seeded on glass or on fibronectin. Bars indicate mean ± SEM ($n = 4$ experiments; unpaired $t$ tests: #, $P < 0.0001$). **(E)** Plots show single-cell trajectories of 690.cl2 cells after transfection with control (siCtr) and two individual Cdc42EP5 (siEP5#1 and siEP5#3) siRNAs on 2D surfaces. Violin plot shows speed of individual cells. $n$, individual cells; Kruskal–Wallis and Dunn's tests: *, $P < 0.05$; #, $P < 0.0001$. **(F)** Western blots show pY438-Src and tubulin levels in 690.cl2 cells after transfection with control (siCtr) and siRNAs targeting individual Borg genes (siEp1–5). Graph shows the quantification of normalized pY418-Src levels. Bars indicate mean ± SEM ($n = 5$ experiments; one-way paired ANOVA, Dunnett's test: †, not significant; ***, $P < 0.001$). AU, arbitrary units.

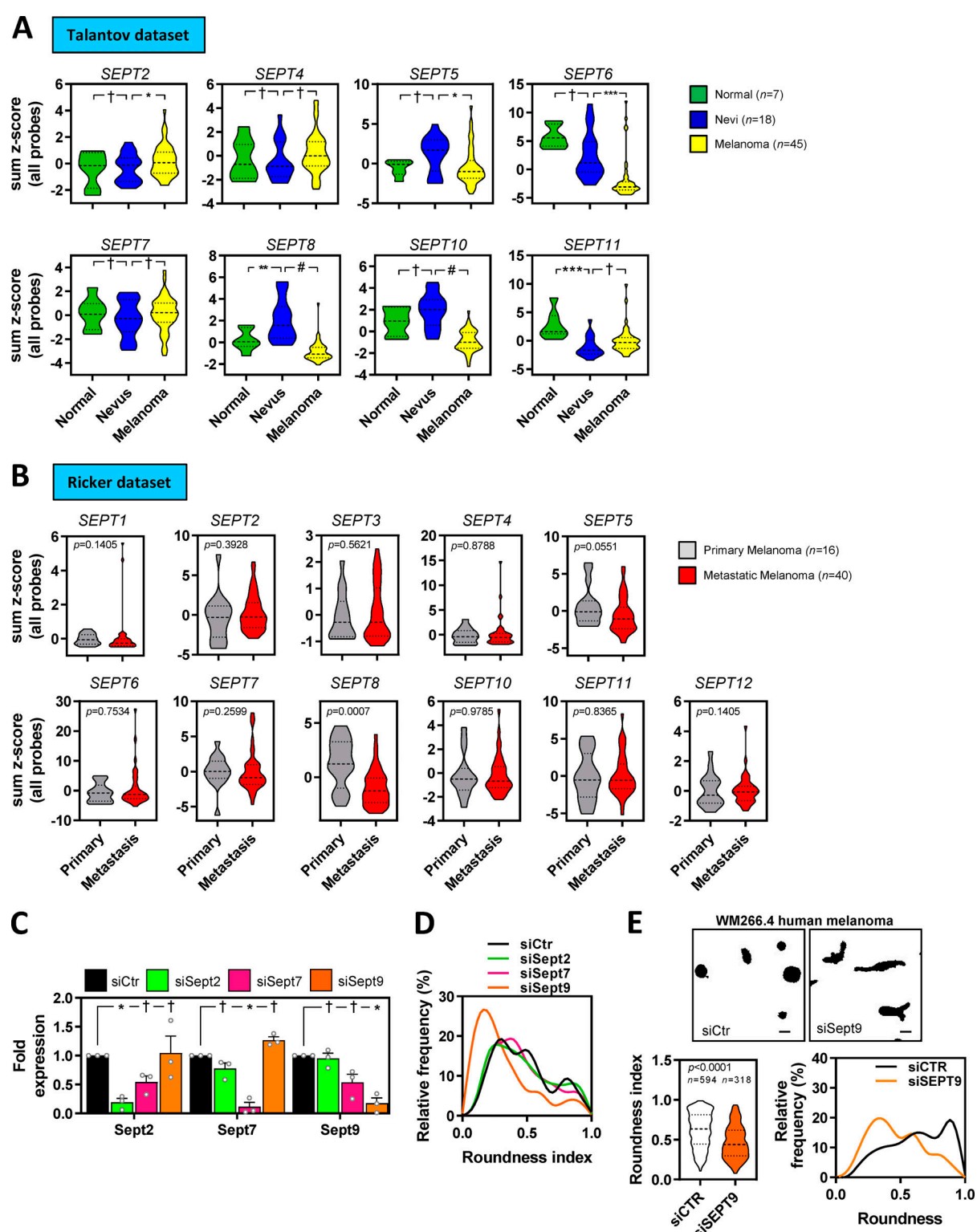

Figure S4. **SEPT9 is associated with melanoma aggressiveness. (A and B)** Violin plots showing the expression of available septin genes in human tissues from normal skin, nevus, and melanoma from the Talantov et al. dataset (GSE3189; A) and primary melanoma and metastatic melanoma from the Riker et al. dataset (GSE7553; B). *n*, individual patients; Kruskal–Wallis and Dunn's tests (A) and Mann–Whitney tests (B): †, not significant; *, P < 0.05; **, P < 0.01; ***, P < 0.001; #, P < 0.0001. **(C)** Graph showing quantification of normalized protein expression levels of Sept2, Sept7, and Sept9 in 690.cl2 cells after transfection with control (siCtr) or Sept2, Sept7, or Sept9 siRNAs. Bars indicate mean ± SEM (*n* = 3 experiments; two-way ANOVA Dunnett's test: †, not significant; *, P < 0.05). Representative Western blot is shown in Fig. 4 B. **(D)** Graph shows the relative frequencies of roundness indexes in 690.cl2 cells after transfection with control (siCtr) or Sept2, Sept7, or Sept9 siRNA on collagen-rich matrices (additional representation of Fig. 4 D). **(E)** Top panels show morphologies of WM266.4 after transfection with control (siCtr) or SEPT9 (siSEPT9) siRNA on top of collagen-rich matrices. Scale bars, 20 µm. Left bottom graph shows roundness index quantification of individual cells (*n* indicates individual cells; Mann–Whitney test). Right bottom graph shows the relative frequencies of roundness indexes.

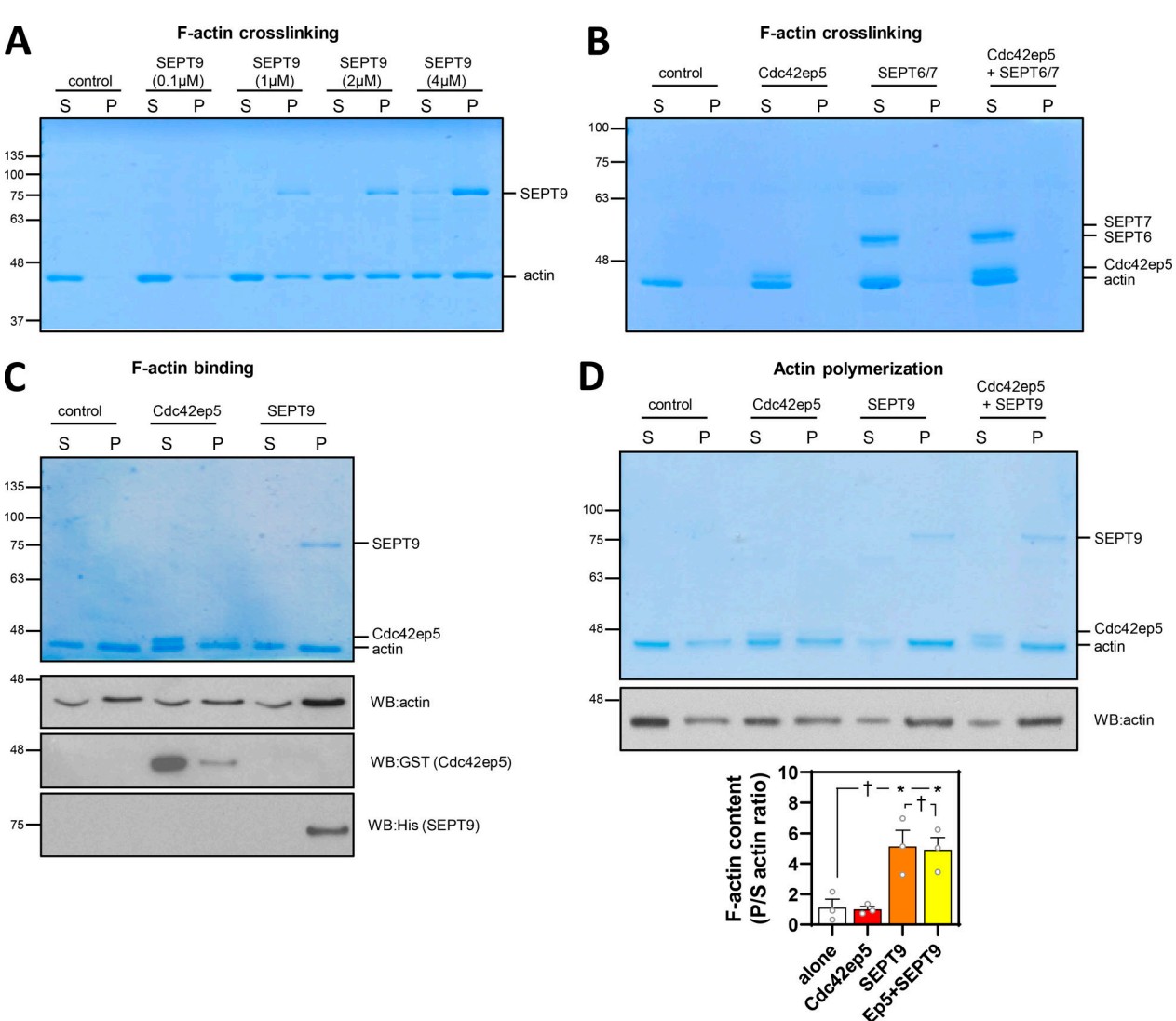

Figure S5. **CDC42EP5 potentiates the F-actin bundling activities of SEPT9. (A)** Coomassie-stained gel showing equal volumes of supernatant (S) and pellet (P) fractions from low-speed sedimentation of prepolymerized actin filaments in the presence of different concentrations of recombinant SEPT9 (as indicated). **(B)** Coomassie-stained gel showing equal volumes of supernatant (S) and pellet (P) fractions from low-speed sedimentation of prepolymerized actin filaments in the presence of recombinant Cdc42ep5 (1 µM), SEPT6/7 (1 µM), and both Cdc42ep5 (1 µM) and SEPT6/7 (1 µM). **(C)** Top panel is a Coomassie-stained gel showing equal volumes of supernatant (S) and pellet (P) fractions from high-speed sedimentation assays (F-actin–binding assays) in the presence of recombinant Cdc42ep5 (1 µM) and SEPT9 (1 µM). Bottom panel shows anti-actin, anti-GST (Cdc42ep5), and anti-His (SEPT9) Western blots of the same experiment. SEPT9 only appears in the pellet (high interaction with F-actin), whereas Cdc42EP5 is primarily found in the supernatant (low interaction with F-actin). **(D)** Top panel is a Coomassie-stained gel showing equal volumes of supernatant (S) and pellet (P) fractions from an actin polymerization assay in the presence of recombinant Cdc42ep5 (1 µM), SEPT9 (1 µM), and both Cdc42ep5 (1 µM) and SEPT9 (1 µM). Bottom panel shows an anti-Actin Western blot of the same experiment. Graph shows the actin polymerization coefficient (ratio of actin in pellet vs supernatant) in the indicated experimental points. Bars indicate mean ± SEM ($n$ = 3 experiments; one-way ANOVA, Tukey's test: †, not significant; *, $P < 0.05$).

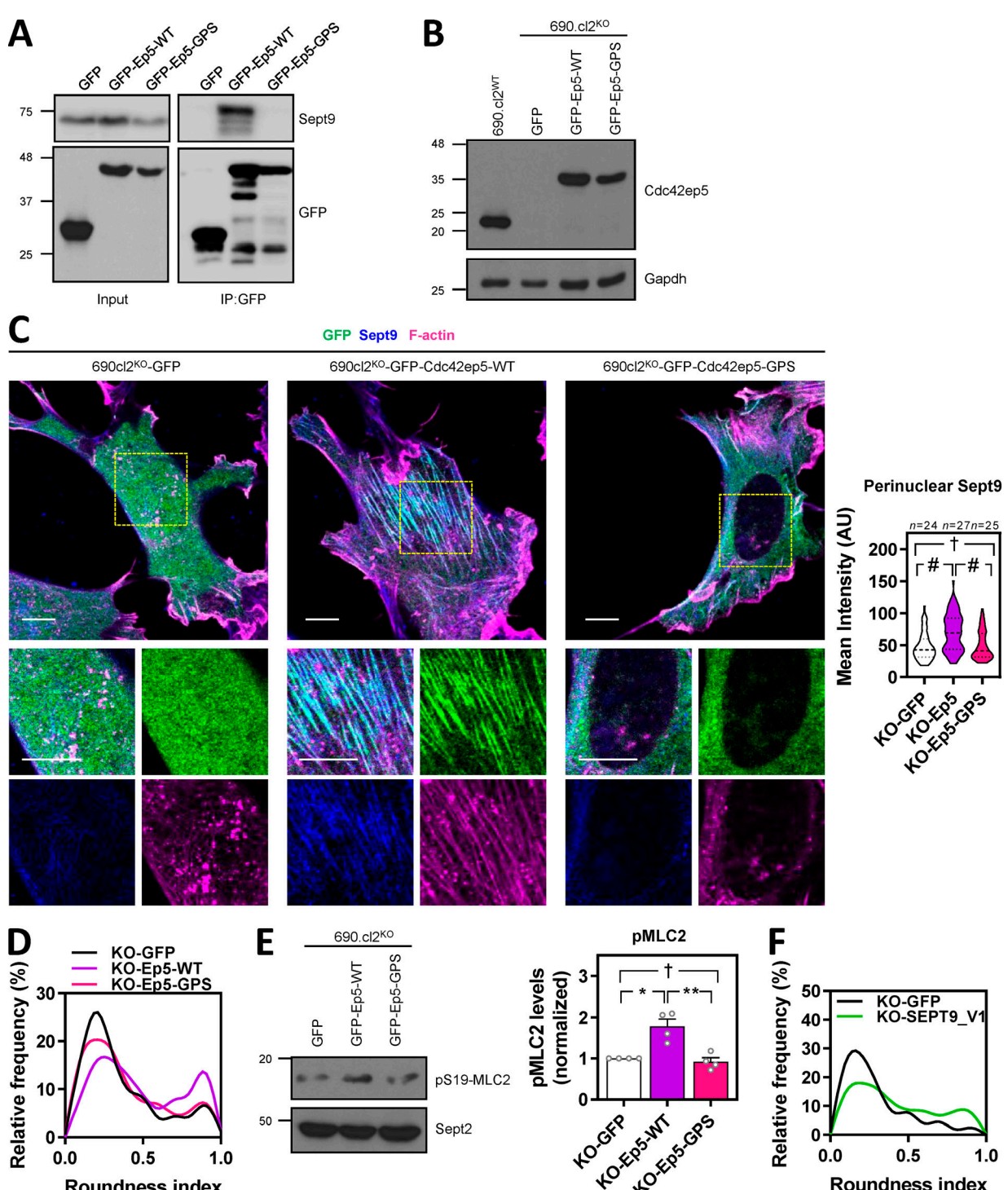

Figure S6. **CDC42EP5 interacts with SEPT9 to promote actomyosin activity. (A)** Western blot showing Sept9 and GFP levels in total lysates (input) and anti-GFP immunoprecipitates (IP:GFP) in 690.cl2 cells ectopically expressing GFP, GFP-Cdc42ep5$^{WT}$, and GFP-Cdc42ep5$^{GPS-AAA}$. **(B)** Western blot showing Cdc42ep5 and Gapdh expression in parental 690.cl2 cells (wild-type) and 690.cl2$^{KO}$ cells expressing GFP, GFP-Cdc42ep5$^{WT}$, or GFP-Cdc42ep5$^{GPS-AAA}$. **(C)** Images show GFP (green), Sept9 (blue), and F-actin (magenta) in 690.cl2$^{KO}$ cells expressing GFP, GFP-Cdc42ep5$^{WT}$, or GFP-Cdc42ep5$^{GPS-AAA}$ on glass. Bottom panels are merged and single-channel magnifications of perinuclear areas. Scale bars, 20 µm. Violin plot in the right shows perinuclear Sept9 mean intensity in the indicated cells. $n$, individual cells; Kruskal–Wallis and Dunn's tests: †, not significant; #, P < 0.0001. **(D)** Graph shows the relative frequencies of roundness indexes in 690.cl2$^{KO}$ cells expressing GFP, GFP-Cdc42ep5$^{WT}$ or GFP-Cdc42ep5$^{GPS-AAA}$ on collagen-rich matrices (additional representation of Fig. 6 C). **(E)** Western blot showing pS19-MLC2 and Sept2 expression in 690.cl2$^{KO}$ cells expressing GFP, GFP-Cdc42ep5$^{WT}$, or GFP-Cdc42ep5$^{GPS-AAA}$. Graph shows the normalized pS19-MLC2 levels. Bars indicate mean ± SEM ($n$ = 3 experiments; one-way ANOVA, Tukey's test: †, not significant; *, P < 0.05; **, P < 0.01). **(F)** Graph shows the relative frequencies of roundness indexes in 690.cl2$^{KO}$ cells expressing GFP or GFP-SEPT9_V1 on collagen-rich matrices (additional representation of Fig. 6 F). AU, arbitrary units; KO, knockout.

Video 1. **Intravital imaging of 690.cl2 cells expressing Cdc42ep5 in a living tumor.** 690.cl2[KO] cells ectopically expressing GFP-Cdc42ep5 (green) were grown subcutaneously and imaged in vivo using two-photon microscopy, which enabled the acquisition of a second harmonic signal (collagen fibers, magenta). Images were acquired every 14 min for a total of 2 h 45 min. Scale bar, 100 µm. Middle panels show magnifications of indicated areas showing fast amoeboid movement into the matrix. Right panels represent motion analysis images of the same area generated by overlaying blue, green, and red images from different time points. Distinct areas of color indicate motile cells, whereas static regions appear white.

Video 2. **Intravital imaging of 690.cl2 lacking Cdc42ep5 in a living tumor.** 690.cl2[KO] cells ectopically expressing GFP (green) were grown subcutaneously and imaged in vivo using two-photon microscopy, which enabled the acquisition of a second harmonic signal (collagen fibers, magenta). Images were acquired every 14 min for a total of 2 h 45 min. Scale bar, 100 µm. Middle panels show magnifications of indicated areas showing slow protrusive movement of cells and elongated morphology. Right panels represent motion analysis images of the same area generated by overlaying blue, green, and red images from different time points. Distinct areas of color indicate motile cells, whereas static regions appear white.

Video 3. **Dynamic association between F-actin and CDC42EP5.** Time-lapse movie of a 690.cl2 cell expressing LIFEACT-RFP (magenta) and GFP-Cdc42ep5 (green). The cell was seeded on glass and imaged every 1 s (60 s total). Right panels show merged and single-channel magnifications (greyscale) of peripheral and perinuclear and peripheral regions.

Video 4. **CDC42EP5 regulates FA dynamics.** Time-lapse movies of a 690.cl2 cell expressing GFP-Paxillin (gray) after transfection with control (siCtr) or Cdc42ep5 (siEP5) siRNA. The cell was seeded on glass and imaged every 30 s (10 min total). Lower panels represent motion analysis images of the same areas generated by overlaying blue, green, and red images from different time points.

**Provided online are three tables. Table S1 describes RNAi, Table S2 describes oligonucleotides, and Table S3 describes antibodies used in this study.**

