## [Peer Review File · The Journal of Cell Biology]

CDC42EP5/BORG3 modulates SEPT9 to promote actomyosin function, migration and invasion

Aaron Farrugia, Javier Rodriguez, Jose Orgaz, Maria Lucas, Victoria Sanz-Moreno, and Fernando Calvo

Corresponding Author(s): Fernando Calvo, Instituto de Biomedicina y Biotecnologia de Cantabria

Review Timeline:

Submission Date:	2019-12-28
Editorial Decision:	2020-02-26
Revision Received:	2020-04-30
Editorial Decision:	2020-05-11
Revision Received:	2020-05-14

Monitoring Editor: Anna Huttenlocher

Scientific Editor: Tim Spencer

Transaction Report:

DOI: <https://doi.org/10.1083/jcb.201912159>

February 26, 2020

Re: JCB manuscript #201912159

Dr. Fernando Calvo
Instituto de Biomedicina y Biotecnología de Cantabria
c/ Albert Einstein 22, PCTCAN
Santander 39011
Spain

Dear Dr. Calvo,

Thank you for submitting your manuscript entitled "CDC42EP5/BORG3 modulates SEPT9 to promote actomyosin function and melanoma invasion and metastasis". We apologize for the extensive delay in providing you with a decision.

In any case, the manuscript has been evaluated by expert reviewers, whose reports are appended below. Unfortunately, after an assessment of the reviewer feedback, our editorial decision is against publication in JCB.

You will see that although reviewer #2 has voiced some enthusiasm for the study, reviewer #1 feels that the manuscript is not suitable for JCB. In addition, both reviewers have raised a number of substantive concerns which preclude further consideration of the paper at this time.

Although your manuscript is intriguing, we feel that the points raised by the reviewers are more substantial than can be addressed in a typical revision period. Therefore, if you wish to expedite publication of the current data, it may be best to pursue publication at another journal.

Given interest in the topic, however, we would be open to an appeal of our decision and resubmission to JCB of a significantly revised and extended manuscript that fully addresses each of the reviewers' concerns in full. Of course, such a resubmission would be subject to further peer-review (hopefully by the same referees). If you would like to resubmit this work to JCB, please contact the journal office to discuss an appeal of this decision or you may submit an appeal directly through our manuscript submission system. Please note that priority and novelty would be reassessed at resubmission.

Regardless of how you choose to proceed, we hope that the comments below will prove constructive as your work progresses. You can contact the journal office with any questions, cellbio@rockefeller.edu or call (212) 327-8588.

Thank you for thinking of JCB as an appropriate place to publish your work.

Sincerely,

Anna Huttenlocher, MD
Monitoring Editor
Journal of Cell Biology

Reviewer #1 (Comments to the Authors (Required)):

Review: "CDC42EP5/Borg3 modulates SEPT9 to promote actomyosin function and melanoma invasion and metastasis"

In this work, the authors investigate the role of cdc42ep proteins in the migration of melanoma cell lines. They investigate, if knockdown of cdc42ep5 changes cell migration in 2D and 3D assays and in live animals, if it changes actin and myosin distribution and myosin phosphorylation and if it changes stressfibre formation and focal adhesion assembly. And they ask, whether these functions are mediated by SEPT9.

Specifically, the authors claim that

1) Cdc42ep5 is required for migration of cancer cells in 2D and 3D transwell assays and in metastasis models in mice.

This claim is mainly supported by the data presented, besides that the movies of migration in animals are not convincing.

2) The authors claim that cdc42ep5 reduces actomyosin contractility.

Since cdc42ep5 knockdown reduces septin assembly and stress fibre formation as the authors have shown before and SEPT2 is a known regulator of myosin phosphorylation, this is not at all surprising, but the authors show indeed in a large number of assays and experiments that cdc42ep5 knockdown indeed strongly influences actomyosin contractility and myosin phosphorylation.

3) The authors claim that cdc42ep5 is required for the maturation of focal adhesions

This notion is supported by numerous pieces of evidence presented by the authors, they demonstrate that less focal adhesions form in the knockdown, that FA downstream signaling molecules are underphosphorylated in the knockdown, cells exhibit reduced migration and that stress fibres are reduced in the knockdown.

4) The authors claim that all these effects are mediated via SEPT9.

The authors use knockdown of SEPT2, SEPT7 and SEPT9 to investigate the above processes again for knockdown of SEPT9 and find that SEPT9 knockdown recapitulates cdc42ep5 knockdown in their hands. The authors also find that knockdown of cdc42ep5 reduces sept9 decorated actin perinuclear fibres in 2D and cortical SEPT9 in 3D culture.

5) The authors claim that "SEPT9 is a crucial effector of Cdc42EP5 function in melanoma" by asking, whether the interaction between the cdc42ep BD3 domain and Septins is required for the formation of septin decorated actin stress fibres, the migration of cells like in the cdc42ep and sept9 knockouts and phosphomyosin levels.

Indeed the binding defect cannot rescue for cdc42ep5, but the conclusion by the authors is not supported by the evidence as no other septins are shown in the assays and it is unclear whether this is a SEPT9-specific effect or the interaction of cdc42ep5 with other septins also plays a role.

Overall, the authors did an amazing amount of work and confirmed a role for cdc42eps and septins

in metastasis in a large number of cell culture models. Unfortunately, the resulting insight is not very far beyond what has been done before in terms of molecular cell biological mechanism on cdc42ep-septin-actin interaction (Calvo et al, cell reports, 2015), the interaction of septins with myosin (Joo et al 2007), stress fibres (Kinoshita et al., Genes Dev 1997, Dev Cell 2002) and focal adhesions (Dolat et al, 2014). While every individual observation in itself is thus highly expected from the literature and previous work from the authors in other cell culture models, taken together, the establishment of the in vivo involvement of cdc42eps is an important and necessary piece of work. Although in the reviewers humble opinion it may be better placed in a cancer journal.

Reviewer #2 (Comments to the Authors (Required)):

This is an interesting and thoroughly documented study of the role of Septin 9 and the Borg protein Cdc42ep5 in cell migration, mostly in melanoma cell lines. The authors test each of the 5 known Borgs and several septins for function by RNAi, and selected Cdc42ep5 and Sept9 based on effect size. They use both in vivo and in vitro assays to assess the impact of RNAi or CRISPR-mediated KO of Cdc42ep5 on invasion and metastasis. They demonstrate that the interaction between these two proteins is essential for their effect on actomyosin organization and contractility. Overall, the study is meticulously quantitative, the data are convincing, and the manuscript provides new insights into the function of one of the Borg proteins.

However, there are a few minor issues that need attention:

- 1) In Figure S1A, the authors show RT-PCR data for the Borg RNAi relative to control, but this provides no information on relative expression levels between the different Borgs. For instance, is Cdc42ep5 expressed at much higher levels than the others? Also, for S1B it would be helpful to show the blot from which these data are derived.
- 2) From Fig 1C it appears that the knockdown of Cdc42ep5 is not very efficient (maybe 50%?), but the effect on transwell migration is quite substantial (about 75% inhibited) - a very similar level of effect to the CRISPR KO cells, which is quite puzzling. One would expect some correlation of effect with protein level, but these data do not show this - something that should at least be discussed.
- 3) On p5, it states that GFP-Cdc42ep5 was found to localize with F-actin on the cortex at areas of blebbing activity (Fig S2C) but I do not see this. The GFP appears to be completely diffuse throughout the cell, with no enrichment at the cell cortex of blebs. Either this conclusion needs to be removed, or better and more convincing data are needed.
- 4) A major conclusion of the manuscript is that Cdc42ep5 promotes actin polymerization by SEPT9, but the data in Fig 5E are not convincing - there is very little if any increase in sedimented actinin the blot. The quantification shows a small increase but the figure is not representative. If the authors want to conclude that this is a primary function of Cdc42ep5, I think that stronger data are required.
- 5) The effect size shown in Fig 6D,E is very small compared to that in Fig 1.
- 6) I feel that focusing in the title on melanoma detracts from the generality of the conclusions. Do the authors believe that the interaction of Cdc42ep5 and Sept9 only occurs in melanoma cell lines, and is only of any biological importance in this specific context? It is up to the authors, but i would suggest that removing the word 'melanoma' from the title would attract a broader readership of cell biologists with interests in the cytoskeleton and cell migration.

REBUTTAL LETTER FOR JCB manuscript #201912159

CDC42EP5/BORG3 modulates SEPT9 to promote actomyosin function, migration and invasion - *Note the change in title*

Overall, we are enthusiastic with the response of the reviewers highlighting the amount of solid work and how meticulous and convincing the presented data is:

Reviewer#1: *"[this manuscript] is an important and necessary piece of work", "the authors did an amazing amount of work..."*.

Reviewer#2: *"This is an interesting and thoroughly documented study..."*, *"Overall, the study is meticulously quantitative, the data are convincing, and the manuscript provides new insights into the function of one of the Borg proteins"*

We are also particularly glad that there were no experimental requests and that our experimental layout and approaches were technically sound.

We acknowledge that the reviewers have raised some criticisms and suggested some areas for improvement, and we are grateful for their thoughtful comments. We have addressed all major criticisms in this Rebuttal Letter, as well as making improvements in the revised manuscript. The most significant of these are:

- Discussion regarding the suitability of intravital movies (Rebuttal Letter)
- Discussion of the novelty of our findings in relation to cytoskeletal modulation (Rebuttal Letter).
- Data and discussion on the BD3-mutant (Rebuttal Letter)
- New data on relative expression of Borg genes in our system (new Figure S1I&J)
- Discussion of size effects in RNAi vs KO (Rebuttal Letter)
- Clearer images to demonstrate *in vitro* effect of Cdc42ep5 on Sept9 actin bundling activity (Figure 5E).

Discussions, argumentations and changes in the revised manuscript are described in detail in this point-by-point Rebuttal Letter. To assist in the revision, changes in the manuscript have been highlighted in yellow.

REVIEWER #1

Overall, we are very glad that Reviewer#1 was satisfied with the experimental layout of our study, and had no suggestions regarding additional experiments or major changes in the manuscript. We acknowledge that there were some concerns regarding novelty that we discuss below.

Review: "CDC42EP5/Borg3 modulates SEPT9 to promote actomyosin function and melanoma invasion and metastasis".

In this work, the authors investigate the role of cdc42ep proteins in the migration of melanoma cell lines. They investigate, if knockdown of cdc42ep5 changes cell migration in 2D and 3D assays and in live animals, if it changes actin and myosin distribution and myosin phosphorylation and if it changes

stress fibre formation and focal adhesion assembly. And they ask, whether these functions are mediated by SEPT9.

Specifically, the authors claim that:

1) Cdc42ep5 is required for migration of cancer cells in 2D and 3D transwell assays and in metastasis models in mice. This claim is mainly supported by the data presented, besides that the movies of migration in animals are not convincing.

We appreciate the reviewer's comments underlying the quality of the data, which is robust enough to support our claims. Regarding 'the movies of migration not being convincing', we are not sure what the exact problem is. We have ample experience in this type of experiments (Ferrari et al, *Nat Commun* 2019, PMID: 30631061; Herraiz et al, *J Natl Cancer Inst* 2015, PMID: 26464464; Sanz-Moreno et al, *Cancer Cell* 2011, PMID: 21840487; Sanz-Moreno et al, *Cell* 2008, PMID: 18984162) and have always used similar analyses and representations. Noteworthy, we have followed the standards in the field for experimentation, analysis and representation of cancer cell motility *in vivo* (e.g. Bayarmagnai et al, *Methods Mol Bio* 2018, PMID: 29525998; Sahai, *Nat Rev Cancer* 2007, PMID: 17891189).

Both the Movies and the derived images/representations in Figure 1I illustrate that cells without Cdc42ep5 have defects in *in vivo* motility. Thus, 690.cl2^{KO}-GFP cells appear mainly static (white in the "motion analysis" image), or present just peripheral activity (i.e. emission of protrusions without moving the cell body). On the contrary, 690.cl2^{KO}-GFP-Cdc42ep5 cells have increased motility and 3 examples of fast moving cells are indicated (by arrows). These moving cells appear as a trail of rounded shapes of different colours (blue-green-red) that correspond to the position of the moving cell over different time-frames. We acknowledge that in the movies there may be additional movement (i.e. they may appear "noisy"); this is a problem associated with this type of experiments and linked to small drifts of the tissue during the imaging sessions. As a reminder, these experiments are performed in living mice under anaesthesia for ~3h and therefore, even under perfect experimental conditions, small drifts are expected. Most of these are corrected during post-processing using the collagen second harmonic signal as "non-moving" structures (or static coordinates). This generally works very well for correcting drifts in the imaged confocal plane, but cannot accurately correct for small drifts in the z-plane, which is likely the cause for the minimal drift observed in the movies. Nevertheless, we have focused our analysis on individual cells clearly showing swift changes in localization over several time intervals (i.e. if there is a drift, it will affect all the field of view or a region). In addition, sections of movies showing massive drifts were discarded. In our opinion, representative movies/images clearly show that only cells with Cdc42ep5 are capable of fast migration *in vivo*.

Nevertheless, if the reviewer could be more specific about the problem, we will gladly address it if possible. Maybe the reviewer is suggesting that we include higher magnifications, clearer movies or different representation/examples.

2) The authors claim that cdc42ep5 reduces actomyosin contractility. Since cdc42ep5 knockdown reduces septin assembly and stress fibre formation as the authors have shown before and SEPT2 is a known regulator of myosin phosphorylation, this is not at all surprising, but the authors show indeed

in a large number of assays and experiments that *cdc42ep5* knockdown indeed strongly influences actomyosin contractility and myosin phosphorylation.

We thank the reviewer for this major comment.

In the first sentence the Reviewer refers to our previous study “Cdc42EP3/BORG2 and Septin Network Enables Mechano-transduction and the Emergence of Cancer-Associated Fibroblasts”, Calvo et al, *Cell Rep* 2015 (PMID: 26711338). Indeed, in that study we demonstrate that the Borg family member Cdc42EP3 can modulate stress fibre formation in fibroblasts, and was required for matrix remodelling and associated tumoral functions in cancer-associated fibroblasts. Importantly, no analysis was performed on Cdc42EP5, and cell motility/migration was not assessed. Furthermore, that study describes that the effect of Cdc42EP3 on actin cytoskeleton was dependent on SEPT2&7. SEPT9 organization and function were not investigated. Nevertheless, Cdc42EP5 has indeed been already associated with septin filament organization by seminal studies by Macara’s team (Joberty et al, *Nat Cell Biol* 2001, PMID: 11584266). Again, these studies focused on SEPT2/7 organization, omitted SEPT9, and did not assess any potential role in stress fibre formation or cell migration. We refer to all these studies in our manuscript.

In this new study, we focused in a different cellular process (i.e. migration/invasion, including amoeboid motility) and observed that neither Cdc42EP3 nor SEPT2&7 were absolutely required for these functions, which rely on a different axis (Cdc42EP5-SEPT9)(Figures 1 and 4).

Regarding the previous links between SEPT2 and myosin phosphorylation, it is important to highlight that our study demonstrates a different mechanism of action. Thus, previous findings involving Septin-dependent control of actomyosin function (Joo et al, *Dev Cell* 2007, PMID: 17981136) were related to the modulation of myosin activity by SEPT2 through direct interaction. In our study, we show that depletion of SEPT2 has no effects on amoeboid migration, myosin phosphorylation and associated processes such as cell rounding (Figure 4). Although septins as a family have already been associated with myosin regulation and stress fibre formation, there is still a lack of deep understanding on the relevance of individual septins in specific processes. Here we demonstrate that for amoeboid cell migration, SEPT2 is dispensable whereas SEPT9 is not, suggesting that in this setting SEPT2-dependent regulation of myosin is not happening or is not important. Furthermore, we provide mechanistic insights into the specific dependence on SEPT9 for actomyosin activity as it is the only septin with known actin-bundling activities (Dolat et al, *J Cell Biol* 2014, PMID: 25349260), which were confirmed in our systems (Figure 5 and Figure S5). Thus, we show that the regulation of myosin activity is associated with the stabilisation/disruption of actin filaments by SEPT9.

Our study also demonstrates that SEPT7 silencing does not affect amoeboid migration (Figure 4). It is important to underline that the formation of septin hexamer/octamer (composed of SEPT2/6/7 and SEPT2/6/7/9, respectively)(Dolat et al, *Biol Chem* 2015, PMID: 24114910) depends on the expression of the essential SEPT7 (Zent et al, *Biol Chem* 2011, PMID: 21824007). Thus, one important corollary of our study is that the canonical septin heteroligomers/filaments are not critical for amoeboid migration. This suggests that SEPT9 is capable of functioning on its own (or via alternative and currently uncharacterised mechanisms) for certain processes. We believe these findings are important in the emerging field of septin biology as they illustrate that depending on the system and the cellular processes, some septins are more important than others. Overall, our study provides important additional information against the simplistic view in which all septins (and Borgs) have

overlapped functions. Given the emerging importance of SEPT9, illustrating for the first time how its activity is regulated by Borg proteins and Cdc42EP5 in particular is also very relevant.

There are more discussions regarding novelty of our findings below (see Rebuttal Letter Point#6, Page 6).

3) The authors claim that *cdc42ep5* is required for the maturation of focal adhesions. This notion is supported by numerous pieces of evidence presented by the authors, they demonstrate that less focal adhesions form in the knockdown, that FA downstream signaling molecules are underphosphorylated in the knockdown, cells exhibit reduced migration and that stress fibres are reduced in the knockdown.

Agreed.

This part of the study aimed at exploring whether the defects in amoeboid phenotypes and cytoskeletal structures in 3D-settings after *Cdc42ep5* silencing were extendable to 2D-culture phenotypes. We acknowledge that this part of the study is less novel as similar effects were already described after perturbation of other Borg members (Liu et al, *Mol Cell Biol* 2014, PMID: 24451259; Calvo et al, *Cell Rep* 2015, PMID: 26711338) or SEPT9 (Dolat et al, *J Cell Biol* 2014, PMID: 25349260). Nevertheless, we believe it was important to assess the role of Cdc42EP5 in these particular processes/functions as part of its thorough characterisation. Furthermore, these data are critical for underlying the relevance of the *Cdc42ep5*-Sept9 axis in modulating not only amoeboid migration but also other types of motility as the ones associated with stress fibre/focal adhesion formation (which are also to a large degree myosin dependent) or directed migration in 2D-substrates. Finally, these data provided important information for the subsequent analyses into the mechanism of action.

4) The authors claim that all these effects are mediated via SEPT9. The authors use knockdown of SEPT2, SEPT7 and SEPT9 to investigate the above processes again for knockdown of SEPT9 and find that SEPT9 knockdown recapitulates *cdc42ep5* knockdown in their hands. The authors also find that knockdown of *cdc42ep5* reduces sept9 decorated actin perinuclear fibres in 2D and cortical SEPT9 in 3D culture.

Indeed, that is a nice comprehensive summary of that part of the study. Noteworthy, knocking-down Sept2 or Sept7 had no effect on migration/invasion, whereas silencing Sept9 significantly reduced those processes. As explained in Point#2, we believe this is the first time that SEPT9 has been associated with rounded/amoeboid migration and actomyosin activity by a mechanism that differs from SEPT2-mediated myosin regulation during cytokinesis (Joo et al, *Dev Cell* 2007, PMID: 17981136).

5) The authors claim that "SEPT9 is a crucial effector of Cdc42EP5 function in melanoma" by asking, whether the interaction between the *cdc42ep* BD3 domain and Septins is required for the formation of septin decorated actin stress fibres, the migration of cells like in the *cdc42ep* and *sept9* knockouts and phosphomyosin levels. Indeed the binding defect cannot rescue for *cdc42ep5*, but the conclusion by the authors is not supported by the evidence as no other septins are shown in the

assays and it is unclear whether this is a SEPT9-specific effect or the interaction of cdc42ep5 with other septins also plays a role.

The reviewer is right in that the BD3 mutant of Cdc42ep5 may be defective in binding to all septins (not just Sept9) and therefore a potential role of other septins needs to be taken into account. This would have been particularly relevant if we had not previously provided strong evidence on the specific role of Sept9 (and not other canonical septins) in cell migration and associated processes (Figure 4), and the effect of Cdc42ep5 in Sept9-dependent actin bundling (Figure 5).

We confirmed that Cdc42ep5 is capable of immune-precipitating endogenous Sept2,7,9, suggesting interaction. This interaction was dependent on an intact BD3 domain – see below (**Figure 1a, Rebuttal Letter**). Due to previous literature, space restrictions and to generate a more straight forward story, we decided to focus our attention on Sept9 in the manuscript. This was mainly based on the following points:

- 1) Joberty et al (*Nat Cell Biol* 2001, PMID: 11584266) already demonstrated that Cdc42EP5/Borg3 could interact with SEPT7 (Cdc10) and SEPT2 (Nedd5) through the BD3 domain. Contrary to wild-type Cdc42EP5/Borg3, the BD3 mutant was not capable of inducing changes in SEPT7 (Cdc10) organization. Since Cdc42EP5 interaction with SEPT2&7 had already been demonstrated, we decided to just show the new interaction with SEPT9 (not analysed in the Joberty study), which we showed was also dependent on an intact BD3 domain (Figure S6A). In addition, the defects of the BD3 mutant on SEPT2/7 were not included (see Figure 1b, rebuttal letter) as they were already described by Joberty et al and we decided to focus in the novel aspects (i.e. SEPT9). To clarify this point, we have made small changes in the text (Page 8, Line 293-297).
- 2) Since we had already demonstrated in Figure 4 that other canonical septins such as Sept2 and Sept7 had not relevance in migration or associated phenotypes, it is therefore highly unlikely that Cdc42ep5 depends on them to fulfil its functions in these particular processes.
- 3) Mechanistically, we show that Sept9 can promote F-actin bundling whereas Sept6/7 cannot (confirming previous results by Dolat et al, *J Cell Biol* 2014, PMID: 25349260), and that Cdc42ep5 is capable of enhancing Sept9 bundling activities (Figure 5E and S5). This particular function is in line with the specific cellular phenotypes (stress fibre/actin cortex) and functions (actomyosin activity, migration/invasion) elicited by Cdc42EP5.

Thus, for the specific functions and phenotypes under study the important axis is Cdc42ep5:Sept9. Nevertheless, we acknowledge that Cdc42EP5 may modulate other septins as well, which may be important for still undetermined cellular processes/functions.

Nevertheless, if the reviewer still feels it is appropriate to include the additional data provided in this rebuttal letter in the manuscript, we would gladly follow his/her recommendations.

Rebuttal Figure 1. Cdc42EP5 modulates septins in 690.c12 cells via the BD3 domain. (a) Western blot showing Sept2, 7, 9 and GFP levels in total lysates (input) and anti-GFP immunoprecipitates (IP:GFP) in 690.c12 cells ectopically expressing GFP, GFP-Cdc42ep5^{WT} and GFP-Cdc42ep5^{GPS-AAA}. **(b)** Images show GFP (green), Sept2 (red) and Sept7 (blue) in 690.c12 Cdc42Eep-knock-out cells expressing either GFP, GFP-Cdc42ep5^{WT} or GFP-Cdc42ep5^{GPS-AAA} and seeded on glass. Right panels are single channel magnifications of the indicated perinuclear areas. Scale bar, 20 μ m. Tukey boxplots showing quantification of Sept2 (top) and Sept7 (bottom) perinuclear intensity

6) Overall, the authors did an amazing amount of work and confirmed a role for cdc42eps and septins in metastasis in a large number of cell culture models. Unfortunately, the resulting insight is not very far beyond what has been done before in terms of molecular cell biological mechanism on cdc42ep-septin-actin interaction (Calvo et al, cell reports, 2015), the interaction of septins with myosin (Joo et al 2007), stress fibres (Kinoshita et al.,Genes Dev 1997, Dev Cell 2002) and focal adhesions (Dolat et al, 2014. While every individual observation in itself is thus highly expected from the literature and previous work from the authors in other cell culture models, taken together, the establishment of the in vivo involvement of cdc42eps is an important and necessary piece of work. Although in the reviewers humble opinion it may be better placed in a cancer journal.

In terms of novelty, Reviewer #1 has raised some concerns. In that sense, we believe that the new (and relevant) insights provided by our study reside in the mechanistic details rather than in the general message. For example, we acknowledge that septins have been previously linked to actomyosin function (Joo et al, *Dev Cell* 2007, PMID: 17981136). We have also previously shown that another Borg family member (i.e. Cdc42EP3) modulates septins and actin (Calvo et al, *Cell Rep* 2015, PMID: 26711338). Noteworthy, these previous studies were focused in the role of a different septin (SEPT2) in other cellular processes (i.e. cell division and cell contractility/ECM remodelling, respectively). We now show that actomyosin function associated to cell migration/invasion is controlled primarily by a different Borg (Cdc42EP5) acting on a specific septin (SEPT9). Moreover, our study also describes a molecular mechanism that differs substantially from previous literature. Studies highlighted by Reviewer #1 (and discussed in our original manuscript) showed that Cdc42EP3 (a different member of the Cdc42EP family) acted as a scaffolding protein promoting the interaction between F-actin and SEPT2/7 filaments required for fibroblast activation. We now show that Cdc42EP5 interaction promotes SEPT9 actin bundling activities independently of SEPT2 and SEPT7, which leads to the stabilization of F-actin structures and increased actomyosin function and amoeboid migration.

Following our argumentation in Point#2 (see above), although the potential role of Cdc42EP5 in actomyosin regulation may not come as a surprise (given the aforementioned studies linking other Borgs to septin regulation, and SEPT2 effects on actomyosin activity), a direct link has previously never described. In addition, the relevance of Cdc42EP5-dependent actomyosin activity in explaining the specific requirements for this protein in cell migration/invasion (not present in other Borg family members) and the underlying mechanism of action are quite substantial and deserve credit. To our knowledge, this is the first time that: (i) all Borg genes have been consistently assessed and compared on a particular function i.e. motility/invasion (Figure 1); (ii) Cdc42EP5 has been linked to actin cytoskeleton regulation and cell motility (Figures 1-3); (iii) a direct regulation of SEPT9 function by a Borg protein has been shown (Figure 5); and (iv) SEPT9 has been linked to actomyosin activity and the regulation of amoeboid migration (Figure 4). Importantly, this function was not shared with other canonical and well-characterised septins such as SEPT2 and 7 (Figure 4). Overall, we believe that the finer details presented in our study are critical in providing new insights into the role of individual septins (and their regulation) in modulating cell migration, that extend beyond the broad generalities that the Reviewer is bringing up.

For us it is also very important that, despite these apparent shortcomings, Reviewer#1 is still confident that “While every individual observation in itself is thus highly expected from the literature and previous work from the authors in other cell culture models, taken together, the establishment of the *in vivo* involvement of cdc42eps is an important and necessary piece of work”.

Regarding Reviewer#1 comment “[this manuscript] may be better placed in a cancer journal” we understand it as a sincere opinion that does not necessarily infer unsuitability for publication at JCB. In fact, we seriously weighted this possibility before submission and eventually decided that the molecular cell biology aspects of the study were more relevant than its links to cancer. In that sense, even though we are a cancer biology orientated lab, for this study we were interested since the beginning in the molecular and cellular aspects determining Cdc42ep/septin mechanism of action within the context of cell migration. We eventually decided to use a melanoma model as it was a highly characterised model that will enable us to study both mesenchymal and amoeboid features and establish connections that could lead to new findings. As a result of using this specific model, we have uncovered new molecular biology on amoeboid migration regulation that may be extensible to immune cells and other cellular systems that present this type of migratory behaviour.

In addition, following Reviewer#2 suggestion, we have made some changes in the Title and Abstract (see below) to reduce the emphasis on cancer/melanoma and broaden the message, as we sincerely believe in the generality of our findings in other systems that rely on actomyosin activity.

REVIEWER #2

We appreciate the overall positive comments from Reviewer#2 including “*This is an interesting and thoroughly documented study ...*” and “*the study is meticulously quantitative, the data are convincing, and the manuscript provides new insights into the function of one of the Borg proteins*”. Furthermore, we are particularly glad that for Reviewer#2 the study was conceptually and technically sound, and that there were only minor issues that needed attention. We have addressed them all in this corrected version of the manuscript and provide point-by-point explanations below.

1) In Figure S1A, the authors show RT-PCR data for the Borg RNAi relative to control, but this provides no information on relative expression levels between the different Borgs. For instance, is *Cdc42ep5* expressed at much higher levels than the others?

Reviewer#2 is right in that data presented in Figure S1A (or Figure 1SG) does not provide relative expression levels of the different Borgs. We agree that this information is very relevant as the specific requirement for *Cdc42ep5* in 690.cl2 cells or other models may result from particular higher expression levels. We thank the reviewer for this comment.

We now provide relative expression analyses of all Borg genes in the models that we have used (Figure S1I). In addition, we also include expression levels in melanoma and breast cancer samples from TCGA (RNA-Seq data, Figure S1J). We decided to include this data as further validation, as RT-PCR values may depend on the efficacy of the specific probes used.

Overall, we observe that *Cdc42ep5/CDC42EP5* has a low expression in general, and we conclude that its specific requirement is not fully determined by its relative abundance compared to other Borg genes (Page 4, Lines 106-109).

Also, for S1B it would be helpful to show the blot from which these data are derived.

The original blots were shown in Figure 1C. Due to space limitations in the main Figure, we decided to move the quantification/graph to the Supplementary Figure. To avoid confusion, we have now moved the quantification to the main Figure (Figure 1C).

2) From Fig 1C it appears that the knockdown of *Cdc42ep5* is not very efficient (maybe 50%?), but the effect on transwell migration is quite substantial (about 75% inhibited) - a very similar level of effect to the CRISPR KO cells, which is quite puzzling. One would expect some correlation of effect with protein level, but these data do not show this - something that should at least be discussed.

We thank the Reviewer#2 for this comment. Indeed, just a ~50% reduction in *Cdc42ep5* expression has significant effects in migration and invasion, in line with the defects observed in F-actin, Sept9, focal adhesions, etc. In CRISPR KO cells similar phenotypic and functional defects were observed, despite 100% reduction of *Cdc42ep5* levels. In our opinion, a correlation of effect:protein level cannot be assumed in general terms, as this may depend on the specific stoichiometry of the reaction/mechanism, robustness of the mechanism/system, emergence of adaptive mechanisms, etc. In our experience, effect sizes between transient silencing (e.g. RNAi) and stable depletion (e.g. CRISPR-KO) cannot be fully correlated, as both approaches come with side effects/caveats that may affect the final output. Thus, whereas you may obtain stable depletion via CRISPR KO, mechanisms of cellular adaptation/clonal selection may emerge over time and affect the phenotype under examination. On the other hand, transient silencing leaves very little time for adaptation and may lead to more profound effects, at least in our experience. The important message is that both approaches (i.e. RNAi and CRISPR-KO) produce similar results: they significantly affect phenotype and function.

For example, it may be the case that *Cdc42ep5* levels are already low (as suggested by the new data presented in Figure S1I&J) and that a minor drop leads to a full collapse of the mechanism. This is just one hypothesis; there may be multiple potential causes for this absence of correlation and we consider it is not realistic to discuss them within the context of the current study.

3) On p5, it states that GFP-Cdc42ep5 was found to localize with F-actin on the cortex at areas of blebbing activity (Fig S2C) but I do not see this. The GFP appears to be completely diffuse throughout the cell, with no enrichment at the cell cortex of blebs. Either this conclusion needs to be removed, or better and more convincing data are needed.

We apologise for the lack of clarity in this part of the text.

We agree that GFP-Cdc42ep5 does not exclusively localize in the cortex, but in our opinion it presents a higher intensity in the cortical region. However, we did not provide proper quantitative data for this observation. In addition, we acknowledge that it is not clear GFP-Cdc42ep5 is enriched in blebbing areas – just in the cortex. For both these reasons we have decided to remove that particular conclusion from the manuscript.

We have removed this sentence and the associated Figure (old Figure S2C):

“Actomyosin contractility at the cellular cortex drives the formation of protrusions known as blebs (25). These blebs have been implicated in enhanced migration thorough complex environments, and have been observed in highly invasive melanoma cells (4). Using high resolution confocal microscopy, GFP-Cdc42ep5 was found to localize with F-actin on the cortex at areas of blebbing activity within the cell (Fig S2C)”.

4) A major conclusion of the manuscript is that Cdc42ep5 promotes actin polymerization by SEPT9, but the data in Fig 5E are not convincing - there is very little if any increase in sedimented actin in the blot. The quantification shows a small increase but the figure is not representative. If the authors want to conclude that this is a primary function of Cdc42ep5, I think that stronger data are required.

We apologise for the lack of clarity in the data presented previously. We agree that the Coomassie-Blue image was not convincing in showing increased F-actin after Cdc42ep5 addition on Sept9. We got the impression that this method was not sensitive enough for the subtle changes we aimed to document – we acknowledge we were having problems with the staining and acquisition of images. As a result, we decided to run WB and stain for actin – we thought that this would also enable for more rigorous quantification of differences. In the manuscript we included a representative image of one of this WB that, in our opinion, showed clear differences between the aforementioned experimental points. Following Reviewer#2's suggestion, we have tried different reagents and approaches to stain and photograph the Coomassie-Blue gel. We now include a new representative image that, in our opinion, shows differences (Figure 5D).

Importantly, even though differences were small, they were significant. Noteworthy, this is an *in vitro* assay with just 2 components (actin and Sept9). Being the proposed mechanism of action of Cdc42ep5 indirect (i.e. via Sept9), we think that a ~2-fold increase is particularly relevant and provides mechanistic explanation for the effects observed in cellular systems.

5) The effect size shown in Fig 6D,E is very small compared to that in Fig 1.

Following our argumentation in Point#2 (see above), we do not believe in comparing or performing correlative analyses of RNAi (Fig 1) vs KO characterisation (Fig 6D&E), for reasons explained above. The important message is that similar significant differences were observed using two alternative

approaches. Furthermore, key findings were further validated in other systems (e.g. WM266.4 and MDA-MB-231-LM2, Fig S1G&H).

6) I feel that focusing in the title on melanoma detracts from the generality of the conclusions. Do the authors believe that the interaction of Cdc42ep5 and Sept9 only occurs in melanoma cell lines, and is only of any biological importance in this specific context? It is up to the authors, but i would suggest that removing the word 'melanoma' from the title would attract a broader readership of cell biologists with interests in the cytoskeleton and cell migration.

We appreciate Reviewer#2's sincere comment. Indeed, the data indicates that Cdc42EP5 function is required in several models for migration and invasion, which underlies the potential generality of our findings regarding Cdc42EP5-SEPT9 to other systems. In addition, important phenotypic analyses were further confirmed in mouse embryonic fibroblasts (MEFs). Following Reviewer#2 suggestions, we have removed the word 'melanoma' from the Title and re-written the Abstract and part of the text to make it more general. We believe this will also help to address Reviewer#1's comment regarding suitability for JCB vs Cancer Journal publication (*see above*).

--

In addition to the changes described above, we have included two additional corrections:

- Figure S1F: we have now included the correct graph showing *CDC42EP5* expression levels in WM266.4 and MDA-MB-231-LM2 cells after transfection with control and RNAi against *CDC42EP5*. The previous version described efficacy of RNAi silencing of Borg genes in WM266.4 and MDA-MB-231-LM2 showing fold normalized expression of CDC42EP1-5 (EP1-5) against control cells (siCtr) cells when individual genes were targeted (siEP1-5).
- Both in the text, figures and figure legends we have updated the description of the MDA-MB-231 cells used in the manuscript. These are a metastatic subline called LM2. This line was correctly described in the Methods section in the old version of the manuscript but not in the text.

May 11, 2020

RE: JCB Manuscript #201912159R-A

Dr. Fernando Calvo
Instituto de Biomedicina y Biotecnología de Cantabria
c/ Albert Einstein 22, PCTCAN
Santander 39011
Spain

Dear Dr. Calvo:

Thank you for submitting your revised manuscript entitled "CDC42EP5/BORG3 modulates SEPT9 to promote actomyosin function, migration and invasion". Your paper has been assessed again by reviewer #2 and we would now be happy to publish your paper in JCB pending final revisions necessary to meet our formatting guidelines (see details below).

A. MANUSCRIPT ORGANIZATION AND FORMATTING:

Full guidelines are available on our Instructions for Authors page, <http://jcb.rupress.org/submission-guidelines#revised>. **Submission of a paper that does not conform to JCB guidelines will delay the acceptance of your manuscript.**

- 1) Text limits: Character count for Articles and Tools is < 40,000, not including spaces. Count includes title page, abstract, introduction, results, discussion, and acknowledgments. Count does not include materials and methods, figure legends, references, tables, or supplemental legends.
- 2) Figure formatting: Scale bars must be present on all microscopy images, including inset magnifications. Molecular weight or nucleic acid size markers must be included on all gel electrophoresis.
- 3) Statistical analysis: Error bars on graphic representations of numerical data must be clearly described in the figure legend. The number of independent data points (n) represented in a graph must be indicated in the legend. Statistical methods should be explained in full in the materials and methods. For figures presenting pooled data the statistical measure should be defined in the figure legends. Please also be sure to indicate the statistical tests used in each of your experiments (both in the figure legend itself and in a separate methods section) as well as the parameters of the test (for example, if you ran a t-test, please indicate if it was one- or two-sided, etc.). Also, if you used parametric tests, please indicate if the data distribution was tested for normality (and if so, how). If not, you must state something to the effect that "Data distribution was assumed to be normal but this was not formally tested."
- 4) Materials and methods: Should be comprehensive and not simply reference a previous publication for details on how an experiment was performed. Please provide full descriptions (at

least in brief) in the text for readers who may not have access to referenced manuscripts. The text should not refer to methods "...as previously described."

5) Please be sure to provide the sequences for all of your primers/oligos and RNAi constructs in the materials and methods. You must also indicate in the methods the source, species, and catalog numbers (where appropriate) for all of your antibodies.

6) Microscope image acquisition: The following information must be provided about the acquisition and processing of images:

a. Make and model of microscope

b. Type, magnification, and numerical aperture of the objective lenses

c. Temperature

d. imaging medium

e. Fluorochromes

f. Camera make and model

g. Acquisition software

h. Any software used for image processing subsequent to data acquisition. Please include details and types of operations involved (e.g., type of deconvolution, 3D reconstitutions, surface or volume rendering, gamma adjustments, etc.).

7) References: There is no limit to the number of references cited in a manuscript. References should be cited parenthetically in the text by author and year of publication. Abbreviate the names of journals according to PubMed.

8) Supplemental materials: There are usually fairly strict limits on the allowable amount of supplemental data. Articles/Tools may generally have up to 5 supplemental figures. At the moment, you are over this limit but we will be able to give you the extra space this time.

Please also note that tables, like figures, should be provided as individual, editable files. A summary of all supplemental material should appear at the end of the Materials and methods section.

9) eTOC summary: A ~40-50 word summary that describes the context and significance of the findings for a general readership should be included on the title page. The statement should be written in the present tense and refer to the work in the third person. It should begin with "First author name(s) et al..." to match our preferred style.

10) Conflict of interest statement: JCB requires inclusion of a statement in the acknowledgements regarding competing financial interests. If no competing financial interests exist, please include the following statement: "The authors declare no competing financial interests." If competing interests are declared, please follow your statement of these competing interests with the following statement: "The authors declare no further competing financial interests."

11) A separate author contribution section is required following the Acknowledgments in all research manuscripts. All authors should be mentioned and designated by their first and middle initials and full surnames. We encourage use of the CRediT nomenclature (<https://casrai.org/credit/>).

12) ORCID IDs: ORCID IDs are unique identifiers allowing researchers to create a record of their various scholarly contributions in a single place. At resubmission of your final files, please consider providing an ORCID ID for as many contributing authors as possible.

B. FINAL FILES:

-- High-resolution figure and video files: See our detailed guidelines for preparing your production-ready images, <http://jcb.rupress.org/fig-vid-guidelines>.

Thank you for this interesting contribution, we look forward to publishing your paper in Journal of Cell Biology.

Sincerely,

Anna Huttenlocher, MD
Monitoring Editor
Journal of Cell Biology

Tim Spencer, PhD
Executive Editor
Journal of Cell Biology

Reviewer #2 (Comments to the Authors (Required)):

The authors have addressed most of my concerns by either modifying or deleting text, and adding new data. Although I am not sure I buy the argument that RNAi and Cas9-mediated KO are so different they cannot be compared, the data are consistent with the conclusions drawn, and I feel that the work is now suitable for publication with no further revision.